# The Digital Brain Bank, an open access platform for post-mortem imaging datasets

Benjamin C Tendler[1]*, Taylor Hanayik[1], Olaf Ansorge[2], Sarah Bangerter-Christensen[2], Gregory S Berns[3], Mads F Bertelsen[4], Katherine L Bryant[1], Sean Foxley[1,5], Martijn P van den Heuvel[6,7], Amy FD Howard[1], Istvan N Huszar[1], Alexandre A Khrapitchev[8], Anna Leonte[2], Paul R Manger[9], Ricarda AL Menke[1], Jeroen Mollink[1], Duncan Mortimer[1], Menuka Pallebage-Gamarallage[2], Lea Roumazeilles[10], Jerome Sallet[10,11], Lianne H Scholtens[6], Connor Scott[2], Adele Smart[1,2], Martin R Turner[1,2], Chaoyue Wang[1], Saad Jbabdi[1†], Rogier B Mars[1,12†], Karla L Miller[1*†]

[1]Wellcome Centre for Integrative Neuroimaging, FMRIB, Nuffield Department of Clinical Neurosciences, University of Oxford, Oxford, United Kingdom; [2]Division of Clinical Neurology, Nuffield Department of Clinical Neurosciences, University of Oxford, Oxford, United Kingdom; [3]Psychology Department, Emory University, Atlanta, United States; [4]Centre for Zoo and Wild Animal Health, Copenhagen Zoo, Frederiksberg, Denmark; [5]Department of Radiology, University of Chicago, Chicago, United States; [6]Department of Complex Trait Genetics, Centre for Neurogenomics and Cognitive Research, Amsterdam Neuroscience, Vrije Universiteit Amsterdam, Amsterdam, Netherlands; [7]Department of Child Psychiatry, Amsterdam Neuroscience, Amsterdam UMC, Vrije Universiteit Amsterdam, Amsterdam, Netherlands; [8]Medical Research Council Oxford Institute for Radiation Oncology, University of Oxford, Oxford, United Kingdom; [9]School of Anatomical Sciences, Faculty of Health Sciences, University of the Witwatersrand, Johannesburg, South Africa; [10]Wellcome Centre for Integrative Neuroimaging, Department of Experimental Psychology, University of Oxford, Oxford, United Kingdom; [11]Stem Cell and Brain Research Institute, Université Lyon 1, INSERM, Bron, France; [12]Donders Institute for Brain, Cognition and Behaviour, Radboud University Nijmegen, Nijmegen, Netherlands

*For correspondence:
benjamin.tendler@ndcn.ox.ac.uk (BCT);
karla.miller@ndcn.ox.ac.uk (KLM)

†These authors contributed equally to this work

**Abstract** Post-mortem magnetic resonance imaging (MRI) provides the opportunity to acquire high-resolution datasets to investigate neuroanatomy and validate the origins of image contrast through microscopy comparisons. We introduce the *Digital Brain Bank* (open.win.ox.ac.uk/Digi-talBrainBank), a data release platform providing open access to curated, multimodal post-mortem neuroimaging datasets. Datasets span three themes—*Digital Neuroanatomist*: datasets for detailed neuroanatomical investigations; *Digital Brain Zoo*: datasets for comparative neuroanatomy; and *Digital Pathologist*: datasets for neuropathology investigations. The first *Digital Brain Bank* data release includes 21 distinctive whole-brain diffusion MRI datasets for structural connectivity investigations, alongside microscopy and complementary MRI modalities. This includes one of the highest-resolution whole-brain human diffusion MRI datasets ever acquired, whole-brain diffusion MRI in fourteen nonhuman primate species, and one of the largest post-mortem whole-brain cohort imaging studies in neurodegeneration. The *Digital Brain Bank* is the culmination of our lab's investment into post-mortem MRI methodology and MRI-microscopy analysis techniques. This manuscript

provides a detailed overview of our work with post-mortem imaging to date, including the development of diffusion MRI methods to image large post-mortem samples, including whole, human brains. Taken together, the *Digital Brain Bank* provides cross-scale, cross-species datasets facilitating the incorporation of post-mortem data into neuroimaging studies.

## Editor's evaluation

This paper describes a new open-access digital brain bank of post-mortem brains that have been scanned with high-resolution, multimodal magnetic resonance imaging and with select datasets accompanied by histological data. This valuable resource can be used to study healthy human brains, pathological human brains, and the brains of other species, opening new opportunities for comparative neuroanatomy and the biological validation of non-invasive neuroimaging signals.

## Introduction

Magnetic resonance imaging (MRI) occupies a unique position in the neuroscience toolkit. In humans, MRI is used at the single-subject level diagnostically and is increasingly deployed at the population level in epidemiology (*Marcus et al., 2007*; *Miller et al., 2016*; *Snoek et al., 2021*; *Van Essen et al., 2013*). MRI is well-established in the context of imaging causal manipulations in experimental organisms ranging from mice (*Denic et al., 2011*; *Thiessen et al., 2013*) to nonhuman primates (*Absinta et al., 2017*; *Klink et al., 2021*) and provides precise measurements in cellular and tissue preparations (*Wilhelm et al., 2012*). This extensive landscape of overlap with the broader neuroscience toolkit creates the potential for MRI to facilitate integration between technologies and investigations. Although MRI hardware and acquisition protocols often need to be tailored to a specific domain, the underlying technology associated with all MRI measurements gives rise to a common set of signal forming mechanisms, facilitating cross-domain comparisons. There are few methods available to neuroscientists that span this breadth of domains.

One challenge to the use of MRI as a bridging technology is the need for common measurements — for example, the same MRI measurements made across multiple species, or MRI and microscopy measurements in the same brain tissue (*Mars et al., 2021*). Post-mortem MRI provides unique opportunities for such common measurements. MRI in post-mortem tissue can be used to identify the origins of image contrast through integration with microscopy (*Keren et al., 2015*; *Langkammer et al., 2012*; *Mollink et al., 2017*), directly addressing concerns over the nonspecificity of MRI signals. In this context, post-mortem MRI data are important because they share common signal forming mechanisms with in vivo MRI and a common tissue state with microscopy, providing a framework for investigation across multiple spatial scales. Post-mortem MRI facilitates comparative anatomy investigations in species that are not traditionally accessible for in vivo imaging (*Berns et al., 2015*; *Bhagwandin et al., 2017*; *Grewal et al., 2020*; *Heuer et al., 2019*), including extinct species (*Berns and Ashwell, 2017*). Long post-mortem scans provide the opportunity to push the boundaries of spatial resolution, providing whole human brain coverage reaching voxel sizes of 100–500 μm (*Edlow et al., 2019*; *Foxley et al., 2016*; *Fritz et al., 2019*; *Weigel et al., 2021*), edging closer to microscopy techniques but benefitting from compatibility with in vivo imaging. As a nondestructive technique, post-mortem MRI enables the examination of tissue microstructure whilst preserving tissue, facilitating repeat MRI measurements with novel contrasts and technologies; and more generally, its integration with tools for post-mortem investigations (e.g., histopathology or proteomics).

In this work, we introduce the *Digital Brain Bank* (open.win.ox.ac.uk/DigitalBrainBank), a data release platform resulting from a decade of post-mortem MRI research at the University of Oxford. The *Digital Brain Bank* provides open access to several post-mortem neuroimaging datasets spanning investigations into human neuroanatomy, cross-species neuroanatomy, and neuropathology. All datasets provide post-mortem MRI, including diffusion MRI, with complementary microscopy data (e.g., immunohistochemistry or PLI) included with some datasets.

Our post-mortem imaging research has been specifically aimed at achieving whole-brain post-mortem MRI to support the investigation of multiple brain systems/regions and long-range connections (*Foxley et al., 2014*; *Miller et al., 2011*; *Miller et al., 2012*), and the first release to the *Digital Brain Bank* contains 21 distinct whole-brain post-mortem MRI datasets, including from whole human brains. All datasets are available to access, and prospective users of the *Digital Brain Bank* can explore

a subset of data directly on the *Digital Brain Bank* website using *Tview:* a bespoke, open-source, and web-based image viewer. *Tview* has been developed for efficient browsing of imaging data spanning drastically different spatial scales, from submicron resolution microscopy to millimeter MRI acquisitions. It enables real-time visualization and interaction (zooming/panning) of both MRI and microscopy images, and with flexible overlays of different modalities.

All datasets uploaded to the *Digital Brain Bank* are associated with researchers at the University of Oxford, or from close collaborators. Limited Derived Outputs from users of *Digital Brain Bank* datasets will also be considered for data upload. The first release to the *Digital Brain Bank* includes data from multiple published projects covering a breadth of neuroimaging research, including whole-brain diffusion MRI in 14 nonhuman primate species (*Bryant et al., 2021*; *Roumazeilles et al., 2020*; *Roumazeilles et al., 2021*), and one of the largest post-mortem whole-brain cohort imaging studies combining whole-brain MRI and microscopy in human neurodegeneration (*Pallebage-Gamarallage et al., 2018*). In addition, we present a previously unpublished project providing one of the highest-resolution whole-brain human diffusion MRI datasets ever acquired (500 µm isotropic resolution). The *Digital Brain Bank* will continue to grow over the coming years, with a number of further datasets already at the early stages of curation (*Howard et al., 2019a*; *Martins-Bach et al., 2021*; *Martins-Bach et al., 2020*; *Wu et al., 2021*).

## Results

The *Digital Brain Bank* is accessible at open.win.ox.ac.uk/DigitalBrainBank. Datasets have been organized into categories reflecting three predominant themes of post-mortem neuroimaging research:

- *Digital Anatomist:* datasets for detailed neuroanatomical investigations.
- *Digital Brain Zoo:* datasets for comparative neuroanatomy.
- *Digital Pathologist*: datasets for neuropathology investigations.

Here, we provide an overview of each theme, with examples from available datasets in the first release to the *Digital Brain Bank*. A brief description of all the datasets provided with the first release, alongside relevant publications, is provided in *Table 1*.

### Digital Anatomist

Datasets within the *Digital Anatomist* provide a new direction for answering fundamental questions in neuroanatomy, through ultra-high resolution MRI data and complementary microscopy within the same sample in humans and model nonhuman species.

The long scan times available in post-mortem MRI affords imaging at ultra-high spatial resolutions, facilitating the delineation of small tissue structures within the human brain, one of the key aims of the *Digital Anatomist*. Often, post-mortem investigations are limited to small sections of excised brain tissue that represent a limited anatomical region. However, our developments in whole-brain post-mortem diffusion imaging (*Foxley et al., 2014*; *McNab et al., 2009*; *Miller et al., 2011*; *Miller et al., 2012*; *Tendler et al., 2020b*) provide the opportunity to investigate structural connectivity and gross neuroanatomy, at scales that are unobtainable in vivo. These developments have culminated in the *Human High-Resolution Diffusion MRI-PLI* dataset, providing one of the highest-resolution whole-brain human diffusion MRI datasets ever acquired (500 µm isotropic resolution), as shown in *Figure 1*. Companion datasets acquired at 1 mm and 2 mm (isotropic) provide a comparison at cutting-edge and conventional in vivo resolutions (*Figure 1a*).

In addition to providing a new insight into human neuroanatomy, these data can be used to inform experimental design and the interpretation of results. Here, the *Human High-Resolution Diffusion MRI-PLI* dataset enables users to identify the resolution required to visualize certain brain structures (*Figure 1a*), and how spatial resolution impacts tractography performance (e.g., overcoming 'gyral bias'—*Figure 1b*; *Cottaar et al., 2021*; *Schilling et al., 2018*). Polarised light imaging (PLI) provides estimates of myelinated fiber orientation (*Axer et al., 2011*), and complementary PLI data acquired in a subset of brain regions (4 µm in-plane) facilitates cross-scale comparisons (*Figure 1b and c*).

A further aim of the *Digital Anatomist* is to perform quantitative validations across modalities, relating MRI to microscopic measures. These kinds of analyses can only be achieved with accurately coregistered data, enabling pixel-wise comparisons across modalities acquired at drastically different spatial resolutions. This potential is most clearly seen in the *Human Callosum MRI-PLI-Histology* dataset, which provides diffusion MRI (400 µm isotropic), alongside complementary PLI (4 µm in-plane), and histology (myelin and astrocytes) (0.25 µm in-plane) in three excised human corpus callosum samples

**Table 1.** Description of all datasets provided in the first release to the *Digital Brain Bank*.

All Structural MRI datasets in the first release were acquired using a balanced SSFP (bSSFP) or T2-weighted sequence, which yields strong gray-white matter contrast in formalin-fixed post-mortem tissue. Diffusion MRI datasets were acquired using a combination of diffusion-weighted steady-state free precession (DW-SSFP) and diffusion-weighted spin-echo (DW-SE) sequences. Full details of the motivation behind the choice of sequences and available contrasts are described in the Discussion. †T2* and magnetic susceptibility maps are currently available in 9 out of 12 ALS brains and all control brains. The remaining datasets were either lost during scanner export, or are of insufficient data quality for public release.

| Category | Name | Contents: MRI | Resolution (MRI) | Contents: Microscopy | Relevant publications |
|---|---|---|---|---|---|
| Digital Anatomist | Human High-Resolution Diffusion MRI-PLI | Whole-brain diffusion MRI, structural MRI, quantitative T1 and T2 maps: – Control human brain: 1× | Diffusion MRI: (500 µm, 1 and 2 mm iso.) Structural MRI: 312.5×312.5×500 µm³ T1 map: (0.75×0.75×1.6 mm³) T2 map: (0.75×0.75×1.6 mm³) | Polarised light imaging (4 µm in-plane) in the anterior commissure, corpus callosum, pons, thalamus, and visual cortex (same brain) | Dataset described in this publication (Methodology in Appendix 1), Diffusion MRI processing described in *Tendler et al., 2020b*, T2 mapping described in *Tendler et al., 2021* |
| Digital Anatomist | Human Callosum MRI-PLI-Histology | Corpus callosum diffusion MRI: – Excised control human corpus callosum samples: 3× | Diffusion MRI: (400 µm iso.) | Polarised light imaging (4 µm in-plane), bright-field microscopy images of immunohistochemistry stains (0.25 µm in-plane) for PLP (myelin) and GFAP (astrocyte) (same human corpus callosum samples) | *Mollink et al., 2017* |
| Digital Brain Zoo | | Whole-brain diffusion MRI and structural MRI (available in brains marked with a *):<br>• Bushbaby (*Galago senegalensis*): 1×<br>• Capuchin monkey (*Sepajus apella*): 1×<br>• Chimpanzee* (*Pan troglodytes*): 2×<br>• Colobus monkey (*Colobus guereza*): 1×<br>• Cotton-Top tamarin (*Saguinus oedipus*): 1×<br>• Golden Lion tamarin (*Leontopithecus rosalia*): 1×<br>• Hamadryas baboon* (*Papio hamadryas*): 1×<br>• Macaque monkey (*Macaca mulatta*): 3×<br>• Mangabey (*Lophocebus albigena*): 1×<br>• Night monkey, (*Aotus lemurinus*): 1×<br>• Ring-tailed lemur (*Lemur catta*): 3×<br>• Saki monkey (*Pithecia pithecia*): 1×<br>• Western Lowland gorilla* (*Gorilla gorilla*): 1×<br>• Woolly monkey (*Lagothrix lagotricha*): 1× | Diffusion MRI:<br>300 µm iso.: Bushbaby, Cotton-Top tamarin & Golden Lion Tamarin<br>400 µm iso: Night monkey<br>500 µm iso: Ring-tailed lemur and Saki monkey<br>600 µm iso: Capuchin monkey, Chimpanzee, Colobus monkey, Hamadryas baboon, Macaque monkey, Mangabey, Western Lowland Gorilla and Woolly Monkey<br>Structural MRI<br>200 µm iso: Western Lowland Gorilla<br>220 µm iso: Hamadryas Baboon<br>0.22×0.22×0.19 mm³: 1× Chimpanzee<br>0.375×0.375×0.40 mm³: 1× Chimpanzee | None | 1× Western Lowland gorilla and 1× Chimpanzee described in *Roumazeilles et al., 2020*, 3× Macaque monkey and 3× Ring-Tailed Lemur described in *Roumazeilles et al., 2021*. Hamadryas baboon, Cotton-Top tamarin and Golden Lion tamarin datasets described in this publication (Methodology in Appendix 1). All other datasets described in *Bryant et al., 2021* |
| Digital Brain Zoo | Marsupials | Whole-brain diffusion MRI and structural MRI:<br>• Tasmanian devil (*Sarcophilus harrisii*): 2×<br>• Thylacine (*Thylacinus cynocephalus*): 2× | Diffusion MRI:<br>1 mm iso: 1× Tasmanian devil<br>1.5 mm iso: 1× Tasmanian devil<br>1.1 mm iso: 1× Thylacine<br>1.0×1.1×0.8 mm³: 1× Thylacine<br>Structural MRI<br>330 µm iso: 1× Tasmanian devil and 1× Thylacine<br>330×330×300 µm³: 1× Tasmanian devil<br>500 µm iso: 1× Thylacine | None | *Berns and Ashwell, 2017* |
| Digital Brain Zoo | Cetaceans | Whole-brain diffusion MRI and structural MRI<br>• Common dolphin (*Delphinus delphis*): 1×<br>• Pantropical dolphin (*Stenella attenuata*): 1× | Diffusion MRI: (1.3 mm iso.) Structural MRI: (640×640×500 µm³) | None | *Berns et al., 2015* |
| Digital Brain Zoo | Carnivora | Whole-brain diffusion MRI and structural MRI: – European wolf (*Canis lupus*): 1× | Diffusion MRI: (600 µm iso.) Structural MRI: (220 µm iso.) | None | Dataset described in this publication (Methodology in Appendix 1) |

*Table 1 continued on next page*

*Table 1 continued*

| Category | Name | Contents: MRI | Resolution (MRI) | Contents: Microscopy | Relevant publications |
|---|---|---|---|---|---|
| Digital Pathologist | *Human ALS MRI-Histology* | Whole-brain diffusion MRI, structural MRI, quantitative T1, T2, and T2* maps, magnetic susceptibility maps (selected brains): <br>• Amyotrophic lateral sclerosis (ALS) human brains: 12× <br>• Control human brains: 3× | Diffusion MRI: (850 µm iso.) Structural MRI: (230–250 µm in-plane; 270–500 µm slice) T1 map: (0.65–1 mm in-plane, 0.90–1.6 mm slice) T2 map: (0.65–1 mm in-plane; 0.90–1.6 mm slice) T2*/magnetic susceptibility maps: (0.5 mm in-plane; 1.1–1.3 mm slice) | Bright-field microscopy immunohistochemistry stains (0.50 µm in-plane, exception pTDP43 – 0.25 µm in-plane): pTDP-43, IBA1 (pan microglia), CD68 (activated microglia/ macrophages), PLP (myelin), SMI-312 (axonal phosphorylated neurofilaments), and ferritin (iron storage, subset of regions) <br>Regions: Anterior cingulate cortex, corpus callosum, hippocampus, primary motor cortex, and visual cortex (same brains). <br>Selected multimodal histology available in two brains (1× ALS and 1× Control), and multiregional PLP (available in 10 out of 12 ALS brains and all control brains, 5–8 regions per brain) in first data release – remaining histology being actively curated. | *Pallebage-Gamarallage et al., 2018*, Magnetic susceptibility and T2* mapping protocol described in *Wang et al., 2020*, Diffusion MRI processing described in *Tendler et al., 2020b*, T2 mapping described in *Tendler et al., 2021* |

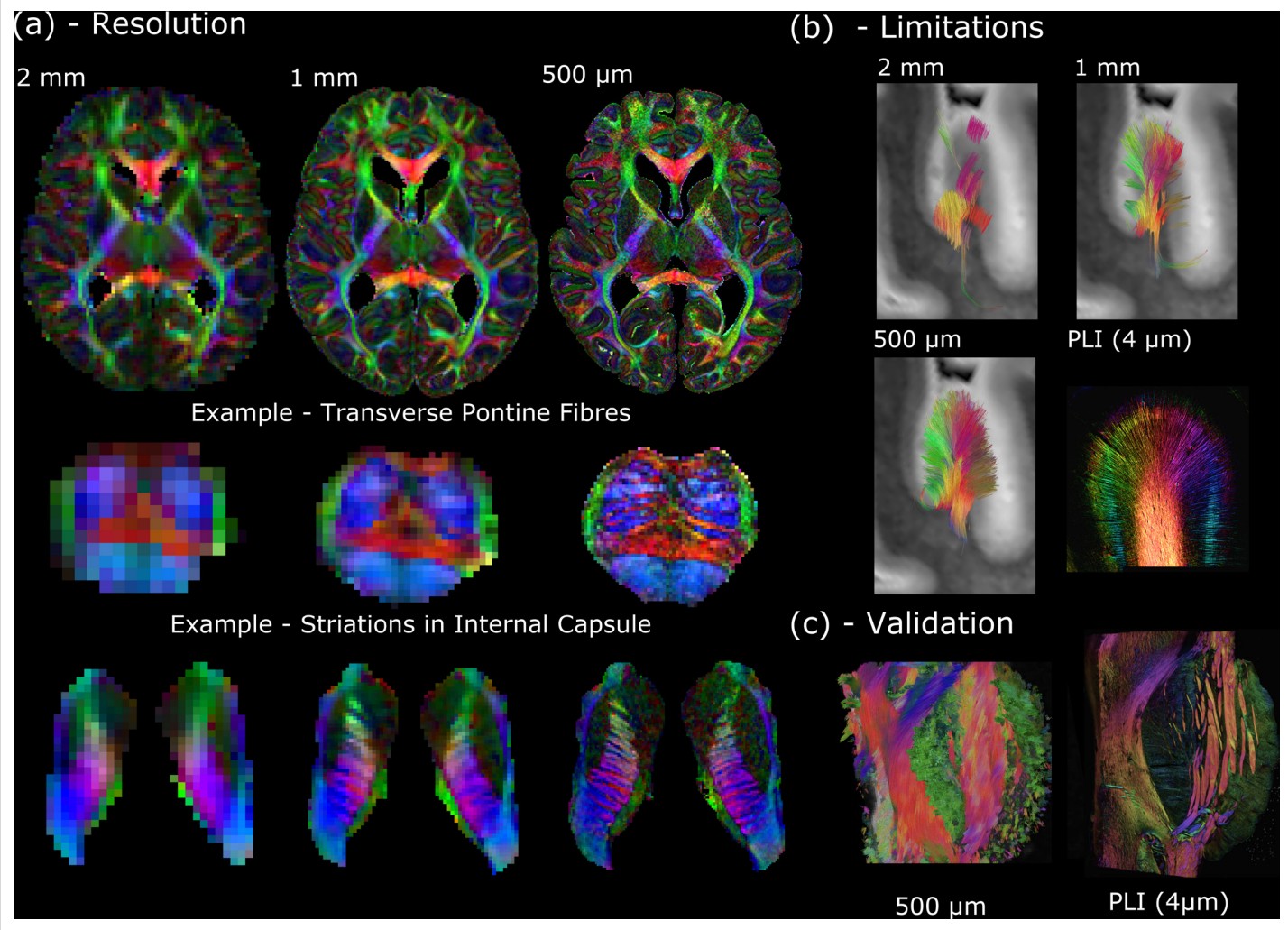

**Figure 1.** The Digital Anatomist. (**a**) Whole-brain diffusion MRI data available in the *Human High-Resolution Diffusion MRI-PLI* dataset reveals the wealth of information provided at increased spatial scales, one of the key aims of the *Digital Anatomist*. Here, the 500 μm dataset uncovers the information hidden at lower spatial resolutions, for example, visualizing the interdigitating transverse pontine fibers with the corticospinal tract or striations through the internal capsule. (**b**) Similarly, datasets across multiple spatial scales can inform us of the limitations at reduced imaging resolutions. Here, gyral tractography (occipital lobe) reveals an overall pattern of fibers turning into the gyral bank at 0.5 mm. At 1 mm, an underestimation of connectivity at the gyral banks is observed, known as the 'gyral bias' (*Cottaar et al., 2021*; *Schilling et al., 2018*). At 2 mm, tractography bears little resemblance to the expected architecture. Multimodal comparisons enable us to validate our findings, with complementary polarised light imaging (PLI) data at over 2 orders of magnitude increase in resolution (125×) revealing a similar pattern of gyral connectivity, and (**c**) excellent visual agreement with tractography across the pons. (**a**) displays diffusion tensor principal diffusion direction maps (modulated by fractional anisotropy).

(*Mollink et al., 2017*). These data offer multiple pathways of investigation, including the identification of the origins of image contrast; validation of microstructural models of tissue (*Mollink et al., 2017*); and developing unique models explicitly linking MRI with microscopy (*Howard et al., 2019b*).

## Digital Brain Zoo

The *Digital Brain Zoo* provides curated datasets to investigate neuroanatomy in nonhuman species and compare anatomy across species.

Post-mortem MRI has enormous potential to inform comparative neuroanatomy for three reasons. First, it enables the scanning of species that would be extremely difficult or impossible to study in vivo. Second, samples can be imaged with minimal handling and without invasive procedures, enabling the study of rare specimens that would not be appropriate to dissect. Third, MRI investigations can be performed in whole-brain samples, rather than excised tissue sections. This makes post-mortem MRI ideally placed to

characterize macroscopic brain structure, long-range structural connectivity, and tissue microstructure in species that are not traditional experimental models, and in particular rare species where very few brain samples may be available (*Berns and Ashwell, 2017*; *Bhagwandin et al., 2017*; *Grewal et al., 2020*; *Mars et al., 2014*).

MRI data from multiple species allows one to formally compare brain organization, important for large-scale comparative neuroscience which has traditionally relied on very limited measures (e.g., whole or regional brain size measures of brain organization) (*Mars et al., 2014*). The ability to acquire data from whole brains opens up the possibility of elucidating *principles* of neural diversity across larger orders of mammalian species (*Friedrich et al., 2021*), and create between-species mappings to formally identify homologies and quantify unique aspects of any given brain (*Mars et al., 2018*). This also allows one to improve translational neuroscience by better understanding the relationship between the human brain and that of model species (e.g., macaque, marmoset, rat, and mouse).

The *Digital Brain Zoo* provides access to post-mortem imaging datasets in nonhuman species covering multiple taxonomic ranks (*Figure 2a*), including nonhuman primate species (*Bryant et al., 2021*; *Roumazeilles et al., 2020*; *Roumazeilles et al., 2021*), Carnivora (*Grewal et al., 2020*), Marsupials (*Berns and Ashwell, 2017*), and Cetaceans (*Berns et al., 2015*). As with other collections in the *Digital Brain Bank*, the *Digital Brain Zoo* currently focuses primarily on whole-brain diffusion MRI. These datasets offer multiple pathways of investigation in comparative neuroanatomy, for example, through the examination of structural connections across brains (*Figure 2b*; *Bryant et al., 2021*). Furthermore, our developments in imaging large post-mortem samples have enabled us to acquire several high-quality post-mortem imaging datasets in species with brains that are too large to fit into specialized preclinical MRI systems, conventionally used to improve image quality in post-mortem MRI (see Discussion).

## Digital Pathologist

Datasets within the *Digital Pathologist* provide a new direction for examining neuropathology and MRI-pathology correlates in humans and established laboratory models.

One of the biggest challenges in the use of MRI clinically is the lack of specificity to disease mechanisms. Many neurological diseases are characterized by changes at the cellular and subcellular level, which cannot be directly visualized with the limited resolution of MRI. Nevertheless, MRI contrast can be made sensitive to cellular-level phenomena that are relevant to disease. Acquisition of MRI and histology in the same tissue enables us to relate microscopic changes in the neural microenvironment to MRI image contrast. The primary aim of the *Digital Pathologist* is to facilitate these cross-scale comparisons, imaging brain tissue associated with a neurological disease.

Such data are provided in the *Human ALS MRI-Histology* dataset (*Figure 3a*), which aims to identify how neuropathological changes in amyotrophic lateral sclerosis (ALS) give rise to altered MRI contrast, and answer specific questions related to ALS pathology. The *Human ALS MRI-Histology* dataset provides whole-brain multimodal MRI and selective histology in a cohort of 12 ALS (diagnosis during lifetime, confirmed ALS neuropathology) and 3 control (no known neuropathology) brains (*Pallebage-Gamarallage et al., 2018*) provided by the Oxford Brain Bank. MRI data includes diffusion, structural, quantitative susceptibility maps (via quantitative susceptibility mapping, QSM), and quantitative T1, T2, and T2* maps. Histology includes markers for proteinopathy (pTDP-43), microglia (CD68 and IBA1), myelin (PLP), neurofilaments (SMI-312), and iron (ferritin) in order to detect changes in a range of microstructures within cortical and subcortical regions (anterior cingulate cortex, corpus callosum, hippocampus, primary motor cortex, and visual cortex) associated with different proposed stages of ALS disease progression (*Jucker and Walker, 2013*).

Different MRI modalities have known sensitivities to different components of the cellular environment. Combined with multimodal histology, these data provide the opportunity to relate neuropathologically induced changes in tissue microstructure to MRI image contrast. While these aims could be partially achieved by dissecting and scanning subregions of the brain, our approach of scanning whole post-mortem brains enables us to investigate neuropathological spread across the entire brain (*Jucker and Walker, 2013*). This facilitates investigations across long-range fiber-tracts associated with pathology (*Figure 3b*), or microstructural changes in multiple brain regions (*Figure 3c*). Notably, these analyses are being facilitated by accurate cross-modality image coregistrations (*Huszar et al., 2019*), enabling us to perform pixel-wise evaluations and integrate structural analyses to identify how pathology influences MR image contrast (*Figure 3d*) in a subset of brain regions associated with

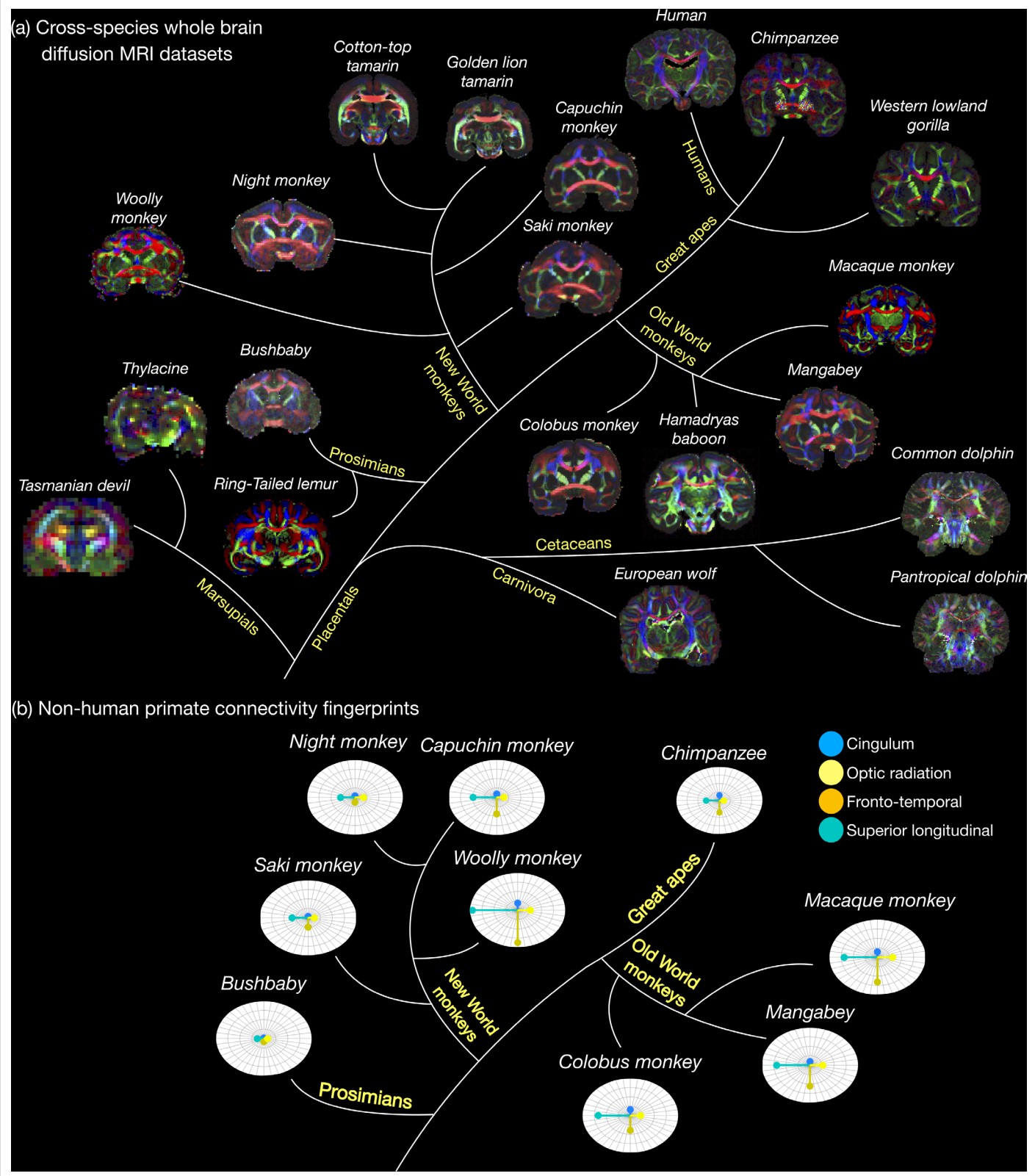

**Figure 2.** The Digital Brain Zoo. (a) The first release of the *Digital Brain Zoo* provides whole-brain MRI datasets spanning multiple species and taxonomic ranks. Notably, we provide whole-brain diffusion MRI datasets from 14 nonhuman primate species, with samples selected for their high quality and to ensure sampling of all major branches of the primate evolutionary tree (Prosimian, New World monkey, Old World monkey, and Great Ape). (**b**) compares the relative volume of four tracts derived from nine nonhuman primate post-mortem datasets provided in the *Digital Brain Zoo* (***Bryant et al., 2021***), where increased distance from the centre corresponds to an increased volume.

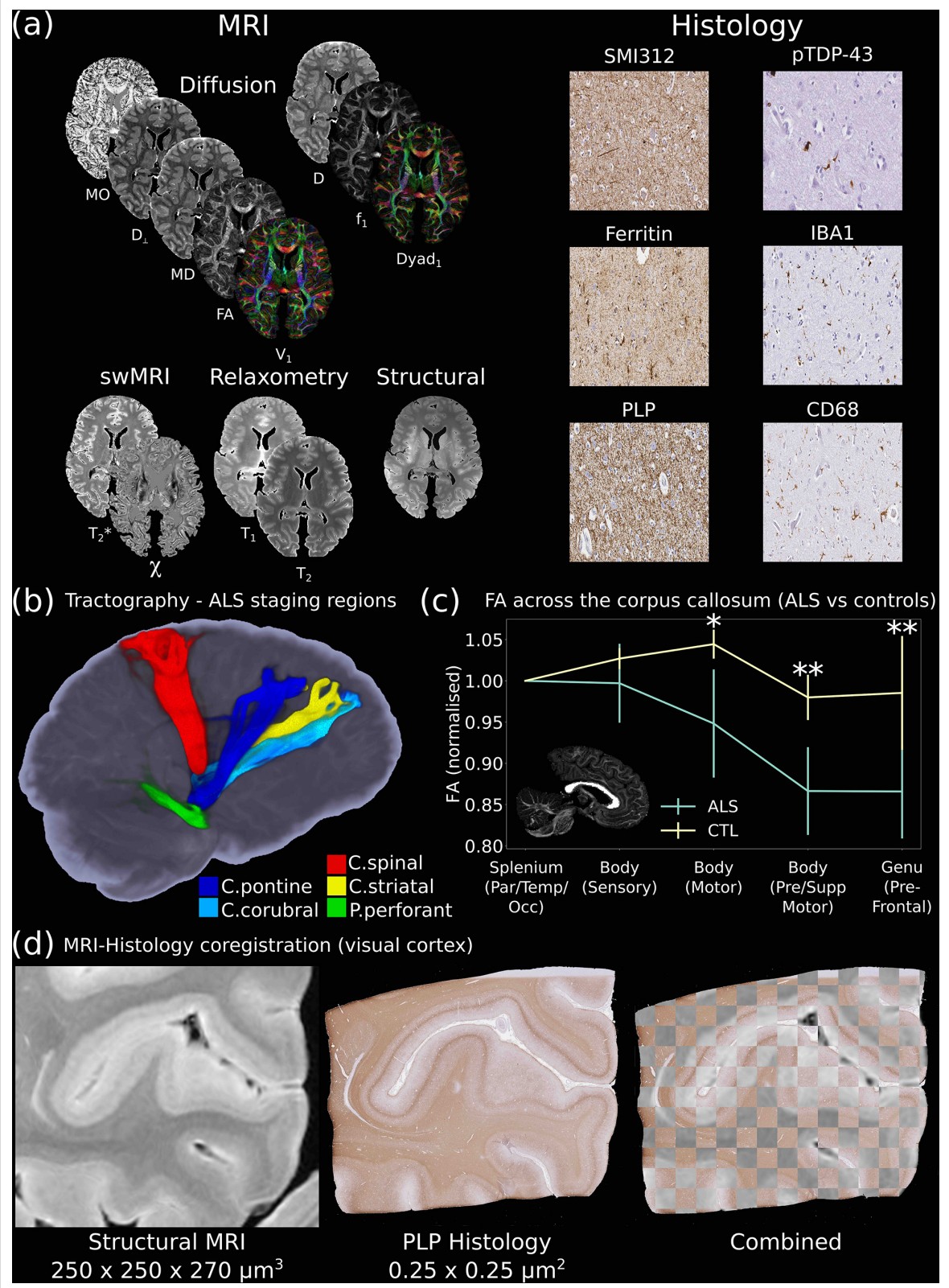

**Figure 3.** The Digital Pathologist. One of the key aims of the *Digital Pathologist* is the examination of neuropathological spread in neurological disease. The *Human ALS MRI-Histology* dataset (**a**) facilitates these investigations, combining whole-brain multimodal MRI and histology (selected brain regions) in a cohort of 12 ALS and 3 control brains. (**b**) Displays the reconstruction of five white matter pathways associated with different ALS stages in a single post-mortem brain (***Kassubek et al., 2014***). Comparisons between ALS and control brains over the corpus callosum of the cohort (**c**) reveals changes

*Figure 3 continued on next page*

Figure 3 continued

in fractional anisotropy (FA, normalized to Par/Temp/Occ lobe), with biggest changes associated with motor and prefrontal regions (*Hofer and Frahm, 2006*) (*=p<0.05; **=p<0.05 following multiple comparison correction) (full details of the corpus callosum analysis provided in Appendix 2). This reflects the anticipated changes in ALS with brain regions associated with motor function, in good agreement with a previous study (*Chapman et al., 2014*), which identified the greatest FA difference between ALS and controls in these regions. Accurate MRI-histology coregistrations facilitate cross-modality comparisons, and (**d**) displays an example of MRI-histology coregistration over the visual cortex of a single ALS brain achieved using the Tensor Image Registration Library (TIRL) (*Huszar et al., 2019*). $V_1$=principal diffusion direction, FA=fractional anisotropy, MD=mean diffusivity, $D_\perp$=radial diffusivity, MO=mode from diffusion tensor output, $Dyad_1$=principal dyad orientation, $f_1$=principal fiber fraction and D=diffusivity from Ball and Two-Stick output, swMRI=susceptibility-weighted MRI, $\chi$=magnetic susceptibility. Details of stain contrasts in (**b**) and (**d**) are provided in *Table 1*. ALS, amyotrophic lateral sclerosis; MRI, magnetic resonance imaging.

different proposed stages of ALS disease progression (*Jucker and Walker, 2013*). MRI data for the *Human ALS MRI-Histology* dataset for all 12 ALS and 3 control brains are immediately available to access, alongside a subset of histology data. Remaining histology data and MRI-histology coregistrations are being actively curated for future release to the *Digital Brain Bank*.

## Tview

The *Digital Brain Bank* website enables users to browse a subset of data easily. A key feature of many datasets is that they contain both MRI and microscopy data. Few available viewers, whether downloadable or online, support both MRI and microscopy file formats, creating a barrier of entry for potential users. Moreover, 2D microscopy datasets are extremely high resolution: single images can exceed 10,000,000,000 pixels, running to gigabytes in size.

We aim to provide online viewing of microscopy and MRI data using standard internet browsers, with a viewer that can handle data at very different spatial scales, provide flexible image overlays, and support color visualization for diffusion-derived measures, PLI, and multiple histological counterstains. Unable to identify existing software with these features, we developed a web-based image viewer, *Tview*. *Tview* is based on software originally used to display satellite imagery at multiple elevations, and enables real-time visualization, interaction (zooming/panning), and flexible overlays of different modalities in a single 2D plane of MRI and microscopy data. Visualization of multimodal (i.e., MRI and microscopy) datasets on the *Digital Brain Bank* website is achieved with *Tview*. An example *Tview* implementation is available at open.win.ox. ac.uk/DigitalBrainBank/#/tileviewer, where cross-modality coregistrations were performed using the Tensor Image Registration Library (TIRL) (*Huszar et al., 2019*) and FNIRT (*Andersson et al., 2007*; *Jenkinson et al., 2012*), both available as part of FSL. Code for *Tview*, the website, and server implementation are available at https://git.fmrib.ox.ac.uk/thanayik/dbb.

The benefits of *Tview* are most readily realized with datasets incorporating MRI and microscopy images, enabling visualization of distinct contrasts over multiple spatial scales. However, many datasets provided in the first release to the *Digital Brain Bank* do not contain any microscopy data. For these datasets, a detailed static image is currently used for visualization on the *Digital Brain Bank* website.

## Requirements for data access and referencing datasets

The *Digital Brain Bank* has been designed to minimize the burden on the user to access datasets, within ethical constraints. For many datasets, we have developed conditions of use terms via a material transfer agreement (MTA), which users agree to prior to access. The MTAs are primarily designed to ensure that datasets are used for research/educational purposes, to prevent misuse, and to satisfy funding requirements.

For datasets restricted by MTAs, when possible, a subset of example data (e.g., data from a single subject) is available to download directly on the website. Upon signing the MTA, users will be granted access to the full dataset. MTAs are currently approved by the University of Oxford. This is currently achieved via the email address provided with each dataset on the *Digital Brain Bank* website. We are actively exploring alternatives to streamline this process.

Upon downloading *Digital Brain Bank* datasets, users agree to acknowledge the source of the data in any outputs. Users are asked to cite the original study for any given dataset and the *Digital Brain Bank*. Details of associated publications and citation instructions are available on an information page associated with each dataset on the *Digital Brain Bank* website.

## Discussion

The *Digital Brain Bank* represents one of the most substantial resources of its kind, providing data from 45 brains across all three themes in the first release. It has been specifically designed to cover the breadth of spatial scales and modalities encountered in post-mortem imaging. Features on the website facilitate data discovery, with users able to interact with a subset of available datasets prior to access. The *Digital Brain Bank* is envisioned as a growing resource reflecting a range of post-mortem neuroimaging projects. Alongside the first release, we aim to first bring together datasets that have been accumulated over the past decade at the University of Oxford. Beyond this, the *Digital Brain Bank* will be the primary resource to release new post-mortem imaging datasets associated with both departmental and collaborative projects.

The *Digital Brain Bank* is a comprehensive resource focusing on post-mortem MRI spanning multiple investigative themes in neuroanatomy, neuropathology, and comparative neuroanatomy. Given this broad coverage, we anticipate that datasets provided by the *Digital Brain Bank* will complement existing open-science initiatives in both human and nonhuman neuroimaging. There are several existing resources providing outputs derived from post-mortem data focusing on other domains, including the *Allen Brain Map* (transcriptomics) (https://portal.brain-map.org/), the *BigBrain Project* (histology) (https://bigbrainproject.org/), and databases compiling datasets from multiple sources such as *EBRAINS* (https://ebrains.eu/). Primarily, we foresee the greatest overlap and integration between the *Digital Brain Bank* and existing databases for in vivo and post-mortem MRI. For example, the high-resolution diffusion MRI datasets provided by the *Digital Anatomist* complement the aims of existing studies such as the *Human Connectome Project* (*Van Essen et al., 2013*), providing the opportunity to validate in vivo findings with higher spatial resolution. The *Digital Brain Zoo's* current focus on nonhuman primates complements several existing in vivo and post-mortem MRI databases, including *PRIME-DE* (*Milham et al., 2018*) and the *JMC Primates Brain Imaging Repository* (*Sakai et al., 2018*). The multiple taxonomic ranks covered by the *Digital Brain Zoo* draws direct parallels with resources such as the Brain Catalogue (*Toro et al., 2014*), which provides nonhuman post-mortem MRI datasets for structural investigations; and the mammalian MRI database (*Assaf et al., 2020*), containing diffusion and T2-/T1-weighted scans of 123 different species (datasets available on request as described in *Assaf et al., 2020*). For the *Digital Pathologist*, we anticipate the strongest integration of our datasets with existing in vivo cohort studies in human or animal models of neuropathology. For the *Human ALS MRI-Histology* dataset, this includes multimodal MRI and biofluid biomarker sampling platforms such as the Oxford Study for Biomarkers in Motor Neurone Disease (*Menke et al., 2014*; *Menke et al., 2015*; *Menke et al., 2016*; *Menke et al., 2018*) and the Canadian ALS Neuroimaging Consortium (*Kalra et al., 2020*). All of these comparisons are supported further by the microscopy data available in select *Digital Brain Bank* datasets, providing the opportunity to link MRI contrast to microscopy-derived features across multiple domains.

## Post-mortem MRI

Post-mortem MRI facilitates the noninvasive investigation of brain anatomy, tissue composition, and structural connectivity through the acquisition of high-resolution datasets and subsequent microscopy comparisons. Despite this potential, post-mortem MRI remains a relatively niche approach, in part due to technical challenges and the need for multidisciplinary expertise. In order to provide post-mortem MRI as an experimental technique to neuroscientists in Oxford, we have had to develop a broad range of underpinning technologies, including: (i) pulse sequences that provide high-quality data under the harsh imaging conditions of post-mortem tissue (*McNab et al., 2009*; *Miller et al., 2011*); (ii) analyses that account for the signal formation mechanisms of these sequences (*Tendler et al., 2020a*; *Tendler et al., 2020b*) or properties unique to post-mortem tissue (*Tendler et al., 2021*); (iii) experimental approaches that enable the use of ultra-high field MRI to increase SNR for high-resolution imaging (*Foxley et al., 2014*; *Tendler et al., 2020b*); (iv) development of custom sample holders to maximize SNR and minimize imaging artifacts (*Appendix 3—figure 1* and *Appendix 3—figure 2*); (v) tools for aligning small 2D microscopy images into 3D whole-brain MRI (*Huszar et al., 2019*); (vi) strategies for co-analyzing MRI and microscopy data (*Howard et al., 2019b*; *Mollink et al., 2017*); and (vii) techniques for between-species comparisons (*Eichert et al., 2020*; *Mars et al., 2018*).

The investment of multidisciplinary expertise and effort required to create these datasets will inevitably be a barrier to similar studies elsewhere. The *Digital Brain Bank* makes our data openly available to researchers worldwide to enable a much broader range of investigations. Considerable work has been performed to process images in a manner that users can immediately incorporate into their own

**Table 2.** Acquisition site and MRI scanner associated with all projects in the first release to the *Digital Brain Bank*.

| Category | Dataset(s) | Acquisition location | MRI scanner |
|---|---|---|---|
| *Digital Anatomist* | *Human High-Resolution Diffusion MRI-PLI* | University of Oxford | Siemens 7T Magnetom 32-channel receive/1-channel transmit head coil (Nova Medical) |
| *Digital Anatomist* | *Human Callosum MRI-PLI-Histology* | University of Oxford | 9.4T 160 mm horizontal bore VNMRS preclinical MRI system 100 mm bore gradient insert (Varian Inc) 26 mm ID quadrature birdcage coil (Rapid Biomedical GmbH) |
| *Digital Brain Zoo* | *NonHuman Primates* | University of Oxford | Baboon, Chimpanzee, Gorilla Siemens 7T Magnetom 28-channel receive/1 channel transmit knee coil (QED) All other brains 7T magnet with Agilent Direct-Drive console 72 mm ID quadrature birdcage RF coil (Rapid Biomedical GmbH) |
| *Digital Brain Zoo* | *Marsupials* | Emory University | Siemens 3T Trio 32-channel receive/1-channel transmit head coil |
| *Digital Brain Zoo* | *Cetaceans* | Emory University | 2× Tasmanian devil and 1× Thylacine Siemens 3T Trio 32-channel receive/1-channel transmit head coil 1× Thylacine Bruker 9.4T BioSpec preclinical MR system |
| *Digital Brain Zoo* | *Carnivora* | University of Oxford | Siemens 7T Magnetom 28-channel receive/1 channel transmit knee coil (QED) |
| *Digital Pathologist* | *Human ALS MRI-Histology* | University of Oxford | Siemens 7T Magnetom 32-channel receive/1-channel transmit head coil (Nova Medical) |

analyses (e.g., diffusion tensor and ball and sticks signal models) reducing user burden to develop their own data processing methods. Further details of these outputs and the types of data available are provided in the Materials and methods.

Here, we provide an overview of how we overcame challenges associated with imaging these samples, notably those associated with imaging large post-mortem brains. Datasets were acquired over many years from multiple imaging sites, resulting in evolving experimental setup, acquisition, and processing methods between datasets. To avoid an exhaustive list of different imaging approaches, below we describe the methodology undertaken for acquisitions performed at the *University of Oxford*, where the majority of datasets in the first release were acquired. Details of the acquisition location and scanner used for all datasets are provided in *Table 2*.

## Choice of MRI scanner

Specialized RF coils and imaging gradients facilitate the acquisition of high-resolution, high-SNR post-mortem MRI datasets. Preclinical systems often deliver in this space, notably with powerful gradient sets, and where possible should be adopted for post-mortem imaging. Specifically, post-mortem tissue that has undergone chemical preservation with aldehyde solutions (e.g., formalin) is characterized by short relaxation time constants (T1, T2, and T2*) (*Birkl et al., 2016*; *Birkl et al., 2018*; *Dawe et al., 2009*; *Kamman et al., 1985*; *Nagara et al., 1987*; *Pfefferbaum et al., 2004*; *Shepherd et al., 2009*; *Thelwall et al., 2006*), and low diffusivity (*D'Arceuil et al., 2007*; *Shepherd et al., 2009*; *Sun et al., 2003*; *Sun et al., 2005*; *Thelwall et al., 2006*) when compared to in vivo tissue. Powerful

gradient sets provide rapid signal sampling and strong diffusion weighing, which boosts SNR versus conventional gradients (*Dyrby et al., 2011*; *Roebroeck et al., 2019*) in this environment.

Broadly speaking, post-mortem MRI data provided in the first release to the *Digital Brain Bank* can be categorised into two experimental designs. Small post-mortem samples (e.g., small NHP brains and excised tissue blocks) were scanned using specialized preclinical systems with powerful gradient sets. At the University of Oxford, these scans were performed with either a 7T preclinical system with Agilent DirectDrive console (Agilent Technologies, CA) or a 9.4T 160 mm horizontal bore VNMRS preclinical MRI system equipped with a 100-mm bore gradient insert (Varian Inc, CA).

Whole brains of larger species do not physically fit into these preclinical systems (maximum sample diameter 7–8 cm) and can only be accommodated in human scanners. These systems often have comparatively low gradient strengths, reducing the available SNR. At the University of Oxford, these brains were scanned on a Siemens 7T Magnetom human scanner. Here, we addressed the imaging environment of fixed, post-mortem tissue and comparatively low gradient-strengths by investing in alternative MR sequences to increase SNR. Further details of this are provided below.

## Sample preparation

All brains and tissue samples in the first release of the *Digital Brain Bank* were chemically fixed using aldehyde solutions (e.g., formalin) to prevent decomposition (*D'Arceuil and de Crespigny, 2007*) and minimize deformation during the course of scanning. All fixed nonhuman brains and excised tissue blocks scanned at the University of Oxford were prepared by soaking the samples in phosphate-buffered saline (PBS) prior to scanning, which increases image SNR by raising $T_2$-values closer to those found in vivo (*Shepherd et al., 2009*). This was not performed in whole human brains, as brain size necessitates a soaking time of multiple weeks for the buffer fluid to penetrate throughout tissue (*Dawe et al., 2009*; *Tendler et al., 2021*; *Yong-Hing et al., 2005*) which was incompatible with our experimental design. We note that soaking tissue for an insufficient time can lead to artificial 'boundaries' in resulting images, where PBS has not penetrated into deep tissue (*Miller et al., 2011*).

## Scanning medium and sample holder

Susceptibility artifacts (arising due to air-tissue or air-medium interfaces) can be exacerbated in post-mortem imaging without an appropriate scanning medium. All samples scanned at the University of Oxford were imaged in a proton-free susceptibility-matched fluid (Fomblin LC08, *Solvay Solexis*; or Fluorinert FC-3283, *3M*). The choice of a proton-free fluid means that there is no signal outside of the brain, bringing the additional advantage of minimizing the required field-of-view for any acquisitions, and addressing scaling issues arising from a bright background signal.

For whole human brain imaging, we built a two-stage custom holder (*Appendix 3—figure 1*), which has become the standard for all of our whole-brain human imaging experiments. The holder was designed to fit into a 32-channel receive/1-channel transmit head coil (Nova Medical), securing brains throughout the acquisition, and contains a spherical cavity to minimize field-inhomogeneities across the brain. The holder enables brains to be placed in a consistent position (equivalent to an in vivo supine scan), minimizing variability of $B_0$-orientation dependent effects (e.g., susceptibility anisotropy; *Liu, 2010*), as well as avoiding any potential motion. While motion is clearly less problematic than in vivo, samples must be well-secured, as even small motions can give rise to coregistration challenges and artifacts across the acquisition period (often >24 hr). All human brains were scanned in this holder, with the exception of the *Human High-Resolution Diffusion MRI-PLI* dataset (data acquired prior to holder construction). Full information on this experimental setup is provided in *Wang et al., 2020*.

Large nonhuman brains scanned at the University of Oxford (Gorilla, Chimpanzee, Wolf, and Baboon) were placed inside a 28-channel receive/1-channel transmit knee coil (QED) to boost SNR (smaller distance between sample and the imaging coil). These brains were placed inside a cylindrical brain holder (*Appendix 3—figure 2*), with a cylindrical cavity that is compatible with the shape of the knee coil, and is a shape that minimizes $B_0$ field inhomogeneities. Small nonhuman brains/excised tissue blocks scanned on preclinical systems were placed in simpler containers, for example, syringes filled with fluorinert.

## Structural MRI

Structural MRI enables the delineation of fine tissue structures and cortical surface reconstruction through high contrast, high-resolution imaging datasets. However, the convergence of T1 relaxation

times for gray and white matter in formalin-fixed post-mortem tissue leads poor contrast with conventional T1-weighted structural protocols (*Miller et al., 2011*). All structural MRIs available in the first data release were acquired using either a balanced SSFP (bSSFP) or T2-weighted sequence, which demonstrate excellent gray/white matter contrast in fixed post-mortem tissue. Notably, bSSFP signal forming mechanisms lead to an extremely high SNR-efficiency (even when considering the reduced T1 and T2 of post-mortem tissue), affording the acquisition of ultra-high resolution (<500 µm) imaging volumes to delineate fine tissue structures in large post-mortem samples.

Contrast in bSSFP and T2-weighted structural MRI datasets is reversed in comparison to conventional in vivo T1-weighted acquisitions (i.e., gray matter appears bright, and white matter appears dark). For these datasets, image contrast is predominantly driven by gray and white matter, facilitating the delineation of fine tissue structures and surface reconstructions (*Roumazeilles et al., 2020*). An example bSSFP dataset is displayed in *Appendix 4—figure 1*. Integration with conventional structural MRI processing pipelines often needs to account for the reversal of image contrast.

## Diffusion MRI

Post-mortem diffusion MRI is particularly challenging due to the MR-relevant properties of fixed tissue, with reductions in measured relaxation time constants T1, T2, and T2* (*Birkl et al., 2016*; *Birkl et al., 2018*; *Dawe et al., 2009*; *Kamman et al., 1985*; *Nagara et al., 1987*; *Pfefferbaum et al., 2004*; *Shepherd et al., 2009*; *Thelwall et al., 2006*), and diffusivity (*D'Arceuil et al., 2007*; *Shepherd et al., 2009*; *Sun et al., 2003*; *Sun et al., 2005*; *Thelwall et al., 2006*) routinely reported in literature.

To achieve high SNR in these conditions, specialized preclinical systems and tissue preparation methods are often required (*Roebroeck et al., 2019*), with many groups focusing on tissue sections that can be scanned on rodent scanners with specialized hardware (*Beaujoin et al., 2018*; *Calabrese et al., 2015*). Unfortunately, size constraints restrict the use of preclinical systems to small post-mortem tissue samples (e.g., small NHP brains or excised tissue blocks). As described above, large whole brains do not physically fit into preclinical systems and can only be accommodated in human scanners. These systems often have comparatively low gradient strengths; combined with conventional methods (e.g., diffusion-weighted spin-echo, DW-SE), this can lead to low-SNR diffusion imaging volumes.

Over the past decade, our lab has invested considerably into the use of an alternative diffusion imaging technique, diffusion-weighted steady-state free precession (DW-SSFP) (*Kaiser et al., 1974*; *Le Bihan, 1988*; *Merboldt et al., 1989a*; *Merboldt et al., 1989b*), to achieve high-SNR datasets in large post-mortem samples. DW-SSFP is well suited to the environment of fixed post-mortem tissue, achieving strong diffusion weighting and rapid signal sampling, even when hardware achieves limited gradient amplitudes and when $T_2$ values are low (*McNab et al., 2009*; *Vasung et al., 2019*; *Wilkinson et al., 2016*). The DW-SSFP sequence has demonstrated improved SNR-efficiency compared to conventional DW-SE when imaging post-mortem tissue (*Miller et al., 2012*), further enhanced at ultra-high field (7T) (*Foxley et al., 2014*). For more details regarding DW-SSFP, please see *McNab and Miller, 2010*.

Broadly, two separate diffusion imaging approaches were used for the first release of data to the *Digital Brain Bank*. Small brains and excised tissue blocks imaged on preclinical systems were scanned using conventional DW-SE sequences, where tissue preparation and powerful diffusion gradients provide imaging volumes with high SNR. Diffusion imaging for larger post-mortem samples scanned on a human scanner (Siemens 7T Magnetom) was performed using DW-SSFP.

To facilitate cross-dataset comparisons, the majority of diffusion datasets from the *Digital Brain Bank* provide derived diffusivity estimates in the form of diffusion tensor and/or ball and sticks model parameters (*Behrens et al., 2007*). Whilst there are a number of standard software packages available for DW-SE data, this was achieved for DW-SSFP datasets using a custom imaging pipelines incorporating the full DW-SSFP model (including $T_1$, $T_2$, and $B_1$ dependencies) (*Tendler et al., 2020b*).

There are some differences between derived diffusivity estimates from DW-SSFP and DW-SE data. Importantly, the DW-SSFP signal does not have a well-defined b-value (*McNab and Miller, 2010*; *Tendler et al., 2020a*). For all DW-SSFP datasets acquired in whole human brains (e.g., the *Human ALS MRI-Histology* dataset and the *Human High-Resolution Diffusion MRI-PLI* dataset), we utilized a recently proposed approach to transform DW-SSFP datasets acquired at two flip angles into equivalent measurements at a single, well-defined b-value (*Tendler et al., 2020a*; *Tendler et al., 2020b*). This facilitates within dataset comparisons, alongside comparisons with datasets acquired with the DW-SE sequence. However, this approach was not possible for the nonhuman DW-SSFP datasets

due to differences in the acquisition protocol. Although the diffusivity estimates for the nonhuman DW-SSFP datasets directly relate to the underlying diffusivity of tissue, the DW-SSFP signal forming mechanisms lead to varying effective b-values within and between these datasets (*Tendler et al., 2020a*). Conservatively, we recommend the nonhuman DW-SSFP datasets to be primarily used for structural connectivity (e.g., tractography) investigations.

More generally, differences in the number of diffusion directions, choice of b-value (for DW-SE and DW-SSFP transformed datasets), imaging resolution, and SNR exist across datasets, a result of available scanning hardware, scanning time, experimental design, and sample properties (e.g., type of fixative used and size of the brain). These limitations can lead to differences in resulting diffusivity estimates and should be considered when performing comparisons across different datasets in the *Digital Brain Bank*. Full details of the acquisitions are provided in the original publications, alongside information on the *Digital Brain Bank* website and dataset files.

## Other sequences

Quantitative T1 and T2 maps are provided with the post-mortem whole-brain human datasets, acquired using conventional turbo inversion-recovery (TIR) and turbo spin-echo (TSE) sequences. Notably, T1-convergence of gray and white matter in fixed post-mortem tissue leads to low contrast on T1 maps, as described in the *Structural MRI* section above. T1 maps were fitted assuming mono-exponential signal recovery. T2 maps were processed using an extended phase graph (EPG) fitting scheme (*Weigel, 2015*), which accounts for B1-inhomogeneity at 7T (details of acquisition and processing are described in *Tendler et al., 2021*).

Whole human brain quantitative T2* and quantitative susceptibility maps are available in a subset of brains provided with the *Human ALS MRI-Histology* dataset. These data were acquired using a multiecho gradient-echo sequence and processed following the procedure in *Wang et al., 2020*.

## Cross-scale comparisons

Post-mortem imaging experiments combining MRI and microscopy are routinely used to validate the origins of image contrast. However, these comparisons are often restricted to simple summary statistics (e.g., ROI averages), rather than utilizing all the available data through pixel-wise comparisons and structural analyses (*Mollink et al., 2017*). These more detailed approaches are facilitated by accurate cross-modality coregistrations, a considerable challenge given differences in image contrast and tissue deformations arising from microscopy processing (*Huszar et al., 2019*; *Iglesias et al., 2018*; *Ohnishi et al., 2016*). These challenges are further exacerbated when considering small tissue sections excised from large post-mortem samples, where the corresponding microscopy sampling region must be identified in a 3D imaging volume. To address this, our group has developed TIRL, a novel MR-microscopy coregistration toolbox (*Huszar et al., 2019*). Further details are provided in the Materials and methods.

## Future directions: dataset visualization

To improve visualization of MRI-only datasets on the *Digital Brain Bank* website, we are currently integrating *NiiVue* (*Rorden et al., 2021*), a web-based 3D volumetric viewer for navigating MRI datasets. *NiiVue* additionally supports binary overlays, which will be used to visualize the location of tissue sampling in the brain. Further details are available at https://github.com/niivue/niivue, (copy archived at swh:1:rev:e67273337430a378a41d6753d91364e9e89b4d33, *Hanayik, 2022*).

## Future directions: available datasets

Several datasets are under active preparation for future release to the *Digital Brain Bank*, notably extending the *Digital Anatomist* and *Digital Pathologist* categories beyond human tissue for neuroanatomical and neuropathological investigations. These datasets include — *Digital Anatomist*: (1) *Forget-Me-Not developing Human Connectome Project (dHCP) study* (*Wu et al., 2021*), providing diffusion MRI datasets acquired in unfixed, post-mortem neonatal brains; (2) *BigMac dataset* (*Howard et al., 2019a*) providing in vivo MRI, post-mortem MRI, PLI, and immunohistochemistry in a single, whole macaque brain. *Digital Brain Zoo*: further primate species are currently in preparation, as are extensions into orders Carnivora and Rodentia. *Digital Pathologist*: a cohort study combining multimodal MRI and histology to investigate mouse models of ALS (*Martins-Bach et al., 2020*; *Martins-Bach et al., 2021*).

## Materials and methods
### Web development and Tview
The *Digital Brain Bank* is a web application made up of individual service components, created using a combination of open-source software. Services include the dataset downloader, the website, and *Tview*. The web application is hosted on our own server hardware, and the various services of the application are orchestrated using container management system *Docker* (docker.com).

The core website and user interface were created using *Vue* (vuejs.org), a front-end web framework for composing reusable application components. The *Vue* website communicates with multiple back-end services (via HTTP requests) to retrieve information (e.g., which *Tview* tiles to display, which datasets are available for download). The reactive web application ensures that changes to website are seen in real time, or on the next possible page reload.

*Tview* provides real-time zooming/panning of high-resolution microscopy and MRI overlays using *leafletjs*, a software library originally used to display satellite imagery at multiple elevations. We have adapted our microscopy datasets to be compatible with this software library in order to take advantage of the features it offers. This is achieved by converting images into tiles at multiple zoom levels (via *libvips*), and uploading these tiles to the local web server. Only the relevant subset of tiles is downloaded when users interact with images using *Tview*. Individual tiles comprise a small file size, facilitating real-time interaction.

Continued use of the *Digital Brain Bank* requires a simple process for collaborators to upload post-mortem datasets, and an application administration layer is under active development to facilitate this process. Code for the *Digital Brain Bank* web application is available at https://git.fmrib.ox.ac.uk/thanayik/dbb.

### Datasets
The *Digital Brain Bank* is not designed as a stand-alone resource—when possible, datasets are associated with available publications which extensively describe the methodology used. This approach facilitates the referencing of available datasets, and similarly ensures that sufficient detail is provided on how data were acquired and processed. A list of the associated publications with the first release of datasets is provided in *Table 1*. However, as part of the first release of the *Digital Brain Bank*, we provide a human dataset that has not yet been described in literature, the *Human High-Resolution Diffusion MRI-PLI* dataset. We additionally provide four new species datasets for the *Digital Brain Zoo*, the Hamadryas baboon, Golden Lion Tamarin, Cotton-Top tamarin, and European wolf. A full description of the acquisition and data processing for these data are provided in Appendix 1.

### Tensor Image Registration Library
The *Digital Brain Bank* makes use of the TIRL to perform cross-modality MRI-microscopy coregistrations (*Huszar et al., 2019*). TIRL can be automated for coregistering 3D MR volumes to 2D microscopy images, typically given a set of sequential block-face photographs taken during the tissue dissection process. These coregistrations are available for all released histology in the *Human ALS MRI-Histology* (*Digital Pathologist*) dataset (*Figure 3d*). Remaining coregistrations are being actively curated and will be provided in a future release to the *Digital Brain Bank*.

The decision to present MRI-microscopy coregistrations in the 2D histology space (*Figure 3d*) was chosen to facilitate visualization. During manual histology sampling, the cutting process introduces non-linear deformities. Furthermore, the cutting angle is not constrained to be parallel to the MRI voxel plane. In general, the excised tissue used for histology will pass through multiple MRI voxel planes nonlinearly, limiting visualization in the 3D MRI space. Importantly, TIRL has been specifically designed to deal with the deformations induced during cutting. Our TIRL pipeline has a specific stage for estimating these deformations from photographs and MRI, and a later stage that refines those deformations for the specific histology slice. Further information is provided in *Huszar et al., 2019*.

### Conditions for data uploading
All datasets hosted on the *Digital Brain Bank* are associated with projects performed at the University of Oxford, in collaboration with members of the University of Oxford, or from close collaborators. In addition, limited Derived Outputs from users of *Digital Brain Bank* datasets will also be considered for data upload, subject to quality control on an individual basis. Information regarding the primary contributors to the dataset are explicitly stated on the *Digital Brain Bank* website. All projects must have been granted ethical approval from the relevant brain banks and departmental ethics boards.

Datasets will be shared on the *Digital Brain Bank* website on the condition that data providers do not require co-authorship for any subsequent outputs based on the use of the datasets alone.

## Types of data provided

The *Digital Brain Bank* uses the following definitions:

- Raw Data: MRI and microscopy images obtained directly from the MRI or slide scanner. These data require further processing be useful (i.e., the value in individual pixels is not directly informative about biological processes or properties).
- Pre-Processed Outputs: imaging outputs from individual brains produced by a first stage of processing suitable for a broad range of subsequent analysis. For example, diffusion parameter estimates (e.g., diffusion tensor outputs), quantitative relaxometry maps, or PLI fiber orientation maps. These outputs can be immediately fed into an analysis to answer a hypothesis- or data-driven neuroscientific question.
- Derived Outputs: results produced from subsequent analysis that use Pre-Processed Outputs as an input specific to a domain of neuroscientific investigation. For example, tractography-derived pathway segmentations or group-averaged atlases.

The *Digital Brain Bank* aims to capture a broad range of datasets under the umbrella of post-mortem neuroimaging, and as such we have aimed to keep the resource flexible for uploaded datasets. There are no strict criteria regarding the types and structure of post-mortem data released to the *Digital Brain Bank*. In addition, the *Digital Brain Bank* aims to facilitate the investigation of research hypotheses designed by the user. To reflect this, we primarily provide Pre-Processed Outputs. Limited Derived Outputs associated with specific projects will be also made available at the discretion of the *Digital Brain Bank*. Raw data for a given dataset is available on request.

These pre-processed outputs reduce the burden on the user to develop their own processing pipelines, of particular importance when considering datasets acquired with alternative sequences not addressed with commonly used imaging software (e.g., DW-SSFP) (*Tendler et al., 2020b*), or datasets which required specialized fitting approaches (e.g., EPG fitting required for T2 mapping at 7T, used in the *Human* ALS MRI-Histology and Human High-Resolution Diffusion MRI-PLI datasets) (*Tendler et al., 2021*; *Weigel, 2015*). For diffusion MRI datasets acquired with DW-SE, we also provide the Pre-Processed Outputs of individual diffusion volumes, which users can feed into a broad range of available software. For DW-SSFP, we do not by default provide the individual diffusion volumes, as no widely available diffusion MRI software packages incorporate the DW-SSFP signal model. We share our custom software for analyzing these data through the *Digital Brain Bank*. DW-SSFP diffusion volumes are available upon request, with the caveat that care needs to be taken in analyzing these data in light of the unusual dependences (e.g., T1 and T2) and signal model of DW-SSFP (*Buxton, 1993*; *Tendler et al., 2020a*; *Tendler et al., 2020b*).

First and foremost, the *Digital Brain Bank* is a data sharing resource. Details of the acquisition and processing methodology associated with each dataset is provided with the accompanying manuscripts, on the *Digital Brain Bank* website (Information page associated with each project), and with the downloaded dataset. However, data hosts are encouraged to provide pre-processing code when available. This code will be linked with each dataset on the *Digital Brain Bank* website (on the Information page), or packaged with the dataset download.

For multimodal (MRI and microscopy) datasets in the first release (*Human High-Resolution Diffusion MRI-PLI, Human Callosum MRI-PLI-Histology,* and *Human ALS MRI-Histology* dataset), raw high-resolution microscopy images are provided. A full set of coregistered data to enable MRI-microscopy voxelwise comparisons via TIRL (*Huszar et al., 2019*) are being actively curated for future release (*Figure 3d*). These are currently available for the histology released with the *Human ALS MRI-Histology* dataset.

Metadata specific to the analysis of post-mortem tissue (e.g., fixative type, post-mortem interval, etc.) or relevant to distinguishing individual datasets in a cohort study (e.g., control brain or brain with a neurological disease) is provided when available.

## Data storage database

As is the nature of a data resource associated with both completed and ongoing projects, some datasets will be updated over time. These future releases will typically be associated with new images being made available or improvements to existing processing pipelines. Until more

streamlined data access methods are in place, we will contact users directly to inform them of any updates made to a given dataset. This approach is aligned with the current framework for data access, with users required to contact the Request Data Contact on the *Digital Brain Bank* website to request access.

A future ambition of the *Digital Brain Bank* is to streamline data access procedures by integrating user sign-up, authentication, and approval combined with access to specific dataset versions within our database over time. To achieve this, we are continuously developing the platform and incorporating feedback and feature requests including enabling a programmatic interface to datasets for approved users, and detailed dataset versioning. We will investigate associating dataset versions with DOIs directly on the *Digital Brain Bank* website (or through known providers), to facilitate the tracking and reproducibility of individual datasets and analysis pipelines.

## Acknowledgements

The Digital Brain Bank is supported by the Wellcome Trust (202788/Z/16/Z) and Medical Research Council (MRC, MR/K02213X/1). The Wellcome Centre for Integrative Neuroimaging is supported by core funding from the Wellcome Trust (203139/Z/16/Z). BCT is supported by funding from the Wellcome Trust (202788/Z/16/Z and 222829/Z/21/Z). KLM, AS, and JM are supported by funding from the Wellcome Trust (202788/Z/16/Z), RBM is supported by funding from the Biotechnology and Biological Sciences Research Council (BBSRC) UK (BB/N019814/1) and the Netherlands Organization for Scientific Research NWO (452-13-015), SJ is supported by funding from the Wellcome Trust (221933/Z/20/Z and 215573/Z/19/Z) and the MRC (MR/L009013/1), TH and DM are supported by funding from the Wellcome Centre for Integrative Neuroimaging, OA is supported by funding from the Medical Research Council, Alzheimer's UK, and NIHR Oxford Biomedical Research Centre, MFB is supported by funding from the Alfred Benzon's Foundation, KLB is supported by funding from the Biotechnology and Biological Sciences Research Council (BBSRC) UK (BB/N019814/1), SF and MPG are supported by funding from the MRC (MR/K02213X/1), MPvdH is supported by the Netherlands Organization for Scientific Research NWO (VIDI-452-16-015 and ALW-179) and the European Research Council (ERC-COG 101001062) AFDH and IH are supported by funding from the Engineering and Physical Sciences Research Council (EP/L016052/1) and Medical Research Council (MR/L009013/1), AAK was funded by Cancer Research UK (C5255/A15935), PRM is supported by funding from the National Research Foundation of South Africa, RALM is supported by funding from the Medical Research Council (MR/K01014X/1) and the Wellcome Trust (202788/Z/16/Z), LR is supported by funding from the Biotechnology and Biological Sciences Research Council (BBSRC) UK (BB/M011224/1), JS is supported by funding from the IDEXLYON IMPULSION 2020 (IDEX/IMP/2020/14) and Labex CORTEX (ANR-11-LABX-0042) grant (Université de Lyon), CS is supported by funding from the NIHR Oxford Biomedical Research Centre (BRC), MRT is supported by funding from the Motor Neurone Disease Association, CW is supported by funding from the China Scholarship Council (CSC). Human post-mortem brain datasets for the Digital Anatomist and Digital Pathologist used tissue provided by the Oxford Brain Bank, a research ethics committee (REC) approved, HTA regulated research tissue bank (REC reference 15/SC/0639). The Oxford Brain Bank is supported by the MRC, Brains for Dementia Research (BDR) (Alzheimer Society and Alzheimer Research UK), and the NIHR Oxford Biomedical Research Centre. The views expressed are those of the authors and not necessarily those of the NHS, the NIHR, or the Department of Health. Datasets for the Digital Brain Zoo used tissue provided by the Australian Museum, Copenhagen Zoo, Primate Brain Bank, Save the Tasmanian Devil, Smithsonian, University of Oxford, and Zoological Society of London.

## Additional information

### Competing interests

Saad Jbabdi, Karla L Miller: Reviewing editor, eLife. The other authors declare that no competing interests exist.

## Funding

| Funder | Grant reference number | Author |
| --- | --- | --- |
| Wellcome Trust | 202788/Z/16/Z | Benjamin C Tendler<br>Ricarda AL Menke<br>Jeroen Mollink<br>Adele Smart<br>Karla L Miller |
| Wellcome Trust | 222829/Z/21/Z | Benjamin C Tendler |
| Wellcome Trust | | Taylor Hanayik<br>Duncan Mortimer |
| Medical Research Council, Alzheimer's UK and NIHR Oxford Biomedical Research Centre | | Olaf Ansorge |
| Alfred Benzon's Foundation | | Mads F Bertelsen |
| Biotechnology and Biological Sciences Research Council | BB/N019814/1 | Katherine L Bryant<br>Rogier B Mars |
| Medical Research Council | MR/K02213X/1 | Sean Foxley<br>Menuka Pallebage-Gamarallage |
| Netherlands Organization for Scientific Research NWO | VIDI-452-16-015 | Martijn P van den Heuvel |
| Netherlands Organization for Scientific Research NWO | ALW-179 | Martijn P van den Heuvel |
| European Research Council | ERC-COG 101001062 | Martijn P van den Heuvel |
| Engineering and Physical Sciences Research Council | EP/L016052/1 | Amy FD Howard<br>Istvan N Huszar |
| Medical Research Council | MR/L009013/1 | Istvan N Huszar<br>Amy FD Howard<br>Saad Jbabdi |
| Cancer Research UK | C5255/A15935 | Alexandre A Khrapitchev |
| National Research Foundation of South Africa | | Paul R Manger |
| Medical Research Council | MR/K01014X/1 | Ricarda AL Menke |
| Biotechnology and Biological Sciences Research Council | BB/M011224/1 | Lea Roumazeilles |
| IDEXLYON IMPULSION 2020 | IDEX/IMP/2020/14 | Jerome Sallet |
| Labex CORTEX (Université de Lyon) | ANR-11-LABX-0042 | Jerome Sallet |
| NIHR Oxford Biomedical Research Centre | | Connor Scott |
| Motor Neurone Disease Association | | Martin R Turner |
| China Scholarship Council | | Chaoyue Wang |
| Wellcome Trust | 221933/Z/20/Z | Saad Jbabdi |
| Wellcome Trust | 215573/Z/19/Z | Saad Jbabdi |

| Funder | Grant reference number | Author |
| --- | --- | --- |
| Netherlands Organization for Scientific Research NWO | 452-13-015 | Rogier B Mars |

The funders had no role in study design, data collection and interpretation, or the decision to submit the work for publication.

## Author contributions

Benjamin C Tendler, Led the Digital Brain Bank project, developed processing pipelines for Digital Anatomist (Human High-Resolution Diffusion MRI-PLI), Digital Brain Zoo (Hamadryas baboon, Gorilla, Chimpanzee and European Wolf), and Digital Pathologist (Human ALS MRI-Histology) MRI datasets, contributed to the establishment of MRI acquisition protocols for Digital Brain Zoo (Baboon and European Wolf) datasets, established the Digital Brain Bank website text and figures, contributed to Digital Brain Bank website design, and wrote the manuscript; Taylor Hanayik, Implemented the Digital Brain Bank website, designed and implemented Tview, designed and implemented the website download/upload framework, contributed to the Digital Brain Bank website design and drafted sections of the manuscript; Olaf Ansorge, Contributed to the development the of tissue sampling strategy, neuropathological analysis and recruitment of post-mortem brains tissue for Digital Anatomist (Human High-Resolution Diffusion MRI-PLI, Human Callosum MRI-PLI-Histology) and Digital Pathologist (Human ALS MRI-Histology) datasets; Sarah Bangerter-Christensen, Carried out immunohistochemistry and histology image acquisition that generated the Human ALS MRI-Histology (Digital Pathologist) hippocampus histology dataset; Gregory S Berns, Led the projects which acquired the Marsupial (Tasmanian Devil and Thylacine) and Cetacean (Common and Pantropical Dolphin) datasets (Digital Brain Zoo); Mads F Bertelsen, Acquired Carnivora (European Wolf) and Non-Human Primate (Hamadryas baboon, Cotton-topped tamarin, Golden lion tamarin and Ring-tailed lemur) samples (Digital Brain Zoo); Katherine L Bryant, Led the project which acquired Primate Brain Bank (Bushbaby, Capuchin monkey, Chimpanzee, Colobus monkey, Mangabey, Night monkey, Saki monkey, Woolly monkey) datasets; Sean Foxley, Conceived study design of the Human High-Resolution Diffusion MRI-PLI (Digital Anatomist) dataset, contributed to the establishment of MRI protocols and carried out MRI acquisition for the Human High-Resolution Diffusion MRI-PLI (Digital Anatomist), Non-Human Primate (Gorilla and Chimpanzee) (Digital Brain Zoo), and Human ALS MRI-Histology (Digital Pathologist) datasets, performed the tractography-PLI analysis for the Human High-Resolution Diffusion MRI-PLI (Digital Anatomist) dataset (Figs. 1b and c), and drafted sections of the manuscript; Martijn P van den Heuvel, Contributed to acquisition of resources for the Digital Brain Zoo through the Primate Brain Bank (Bushbaby, Capuchin monkey, Chimpanzee, Colobus monkey, Mangabey, Night monkey, Saki monkey, Woolly monkey); Amy FD Howard, Produced Tview images for the Digital Brain Bank website implementation, provided critical assessment of the Digital Brain Bank website, and contributed to Digital Brain Bank website design; Istvan N Huszar, Established the MRI-microscopy coregistration software (TIRL) and performed the MRI-histology coregistrations for the Human ALS MRI-Histology dataset (Digital Pathologist); Alexandre A Khrapitchev, Established the MRI acquisition protocols and carried out MRI acquisition for the small Non-Human Primate (Bushbaby, Capuchin monkey, Colobus monkey, Cotton-topped tamarin, Golden lion tamarin, Macaque monkey, Mangabey, Night monkey, Ring-tailed lemur, Saki monkey, Woolly monkey) datasets (Digital Brain Zoo); Anna Leonte, Carried out immunohistochemistry and image acquisition for the Human ALS MRI-Histology (Digital Pathologist) anterior cingulate histology dataset; Paul R Manger, Acquired Carnivora (European Wolf) and Non-Human Primate (Hamadryas baboon, Cotton-topped tamarin, Golden lion tamarin, Ring-tailed lemur) samples (Digital Brain Zoo); Ricarda AL Menke, Provided critical assessment of the processing pipelines for Human ALS MRI-Histology dataset (Digital Pathologist); Jeroen Mollink, Led the project which acquired the Human Callosum MRI-PLI-Histology (Digital Anatomist) dataset, and performed the PLI acquisition and analysis for the Human High-Resolution Diffusion MRI-PLI (Digital Anatomist) dataset; Duncan Mortimer, Contributed to the Digital Brain Bank website design and established the data server implementation; Menuka Pallebage-Gamarallage, Led the project which obtained the Human ALS MRI-Histology (Digital Pathologist) histology dataset, establishing protocols and carrying out systematic human brain sampling, immunohistochemistry, image acquisition and analyses of histology datasets. Supported Digital Anatomist projects (Human High-Resolution Diffusion MRI-PLI, Human

Callosum MRI-PLI-Histology) with tissue sampling and histology data generation; Lea Roumazeilles, Led projects which acquired Non-Human Primate (Chimpanzee, Gorilla, Macaque monkey, Ring-tailed lemur) datasets and facilitated the scanning preparation for Digital Brain Zoo samples scanned at the University of Oxford; Jerome Sallet, Facilitated the acquisition and scanning preparation for all Digital Brain Zoo samples scanned at the University of Oxford; Lianne H Scholtens, Selected and prepared samples from the Primate Brain Bank (Bushbaby, Capuchin monkey, Chimpanzee, Colobus monkey, Mangabey, Night monkey, Saki monkey, Woolly monkey) (Digital Brain Zoo); Connor Scott, Supported the recruitment and preparation of human whole brain samples for the Human High-Resolution Diffusion MRI-PLI (Digital Anatomist) and Human ALS MRI-Histology (Digital Pathologist) datasets; Adele Smart, Carried out immunohistochemistry and image acquisition for the Human ALS MRI-Histology (Digital Pathologist) histology dataset; Martin R Turner, Supported the characterisation of the ALS cohort in the Human ALS MRI-Histology (Digital Pathologist) dataset; Chaoyue Wang, Established the analysis pipeline and performed data processing of the T2* and magnetic susceptibility maps for the Human ALS MRI-Histology (Digital Pathologist) dataset; Saad Jbabdi, Established analysis tools for processing diffusion MRI data in the Digital Anatomist (Human High-Resolution Diffusion MRI-PLI), Digital Brain Zoo (Baboon, Chimpanzee, Gorilla and European Wolf), and Digital Pathologist (Human ALS MRI-Histology) datasets, contributed to the Digital Brain Bank website design, and critically appraised the manuscript; Rogier B Mars, Leads the Digital Brain Zoo, led projects associated with the first data release in the Digital Brain Zoo performed at the University of Oxford, contributed to the Digital Brain Bank website design, and critically appraised the manuscript; Karla L Miller, Conceived the Digital Brain Bank, led projects associated with the first data release in the Digital Anatomist and Digital Pathologist, contributed to the establishment of MRI acquisition protocols for Digital Anatomist (Human High-Resolution Diffusion MRI-PLI), Digital Brain Zoo (Hamadryas baboon, Chimpanzee, Gorilla and European Wolf), and Digital Pathologist (Human ALS MRI-Histology) datasets, contributed to the Digital Brain Bank website design, and critically appraised the manuscript

## Author ORCIDs
Benjamin C Tendler http://orcid.org/0000-0003-2095-8665
Katherine L Bryant http://orcid.org/0000-0003-1045-4543
Amy FD Howard http://orcid.org/0000-0003-1154-1913
Alexandre A Khrapitchev http://orcid.org/0000-0002-7616-6635
Duncan Mortimer http://orcid.org/0000-0002-7483-2024
Jerome Sallet http://orcid.org/0000-0002-7878-0209
Connor Scott http://orcid.org/0000-0003-2316-1707
Adele Smart http://orcid.org/0000-0002-4293-5942
Chaoyue Wang http://orcid.org/0000-0001-9402-1563
Karla L Miller http://orcid.org/0000-0002-2511-3189

## Ethics

Human subjects: All human post-mortem datasets described in the first release to the Digital Brain Bank used tissue provided by the Oxford Brain Bank, a research ethics committee (REC) approved, HTA regulated research tissue bank. The studies were conducted under the Oxford Brain Bank's generic Research Ethics Committee approval (15/SC/0639).

There are four new species datasets (Hamadryas baboon, Golden Lion tamarin, Cotton-Top tamarin, and European wolf) provided in the first release to the Digital Brain Bank which have not been previously described in literature. These datasets all used post-mortem tissue from animals which died of causes unrelated to research, and therefore do not require a Home Office license under the Animals (Scientific Procedures) Act 1986. Ethics statements associated with all remaining Digital Brain Bank datasets are described in the original manuscript associated with each dataset, as provided in Table 1.

## Decision letter and Author response
Decision letter https://doi.org/10.7554/eLife.73153.sa1
Author response https://doi.org/10.7554/eLife.73153.sa2

## Additional files

### Supplementary files
• Transparent reporting form
• Supplementary file 1. Corpus Callosum Analysis.

### Data availability
The Digital Brain Bank (https://open.win.ox.ac.uk/DigitalBrainBank) is a data release platform providing open access to curated, multimodal post-mortem neuroimaging datasets. All datasets described in this manuscript are available through the Digital Brain Bank, with details of access provided within the manuscript and on the website. Code for the Digital Brain Bank resource is available at https://git.fmrib.ox.ac.uk/thanayik/dbb. When available, details of associated processing code for each dataset is linked to the dataset's Information page on the Digital Brain Bank website. Source data for the corpus callosum analysis in Fig 3c is provided in a Supplementary File.

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

# Appendix 1

## Human high-resolution diffusion MRI-PLI dataset

### MRI preparation and scanning

Data were acquired from a post-mortem human brain (n=1) with no known neuropathology. The brain was extracted from the skull within 72 hr after death and fixed in 10% PBS buffered formalin (4% formaldehyde) for 6 weeks prior to scanning. The brain was removed from formalin and placed in plastic bags filled with Fomblin LC08 (*Solvay Solexis*), a susceptibility-matched perfluoropolyether liquid that contributes no signal to the imaging experiment.

The brain was imaged with a Siemens 7T whole-body scanner (1 Tx/32 Rx head coil). Diffusion-weighted volumes were acquired using DW-SSFP sequence. As highlighted in the main text, the choice of DW-SSFP was motivated by the sequences potential to simultaneously address the short $T_2$ and low diffusivity of fixed, post-mortem tissue, when limited to human scanners.

Whole-brain diffusion MRI datasets were acquired at 500 μm, 1 mm, and 2 mm isotropic resolution. Details of the acquisition parameters are provided in *Appendix 1—table 1*, where we note that the 500 μm dataset took approximately 6 days of continuous scanning to acquire. DW-SSFP datasets were obtained at two flip angles to address B1-inhomogeneity at 7T, as previously described in *Tendler et al., 2020b*.

The DW-SSFP signal is dependent on tissue relaxation time-constants ($T_1$ and $T_2$) and the acquisition flip angle, which must be estimated for accurate modeling. These parameters were estimated using a TIR, TSE, and actual flip angle imaging (AFI) (*Yarnykh, 2007*) sequence. A structural scan was additionally acquired using a true fast imaging with steady-state precession (TRUFI) sequence (bSSFP), which produces high gray/white matter contrast in post-mortem tissue. Details of the acquisition parameters are provided in *Appendix 1—table 2*.

## MRI processing

A Gibbs ringing correction was applied to the DW-SSFP, TIR, and TSE datasets (*Kellner et al., 2016*). All coregistrations within-and-between modalities were performed using a six degrees-of-freedom coregistration using FSL FLIRT (*Jenkinson and Smith, 2001*). T1 maps were estimated from the TIR volumes assuming mono-exponential signal evolution. T2 maps were estimated from the TSE volumes using an EPG framework (*Weigel, 2015*), as described in *Tendler et al., 2021*. B1 maps were estimated from the AFI volumes as described in the original AFI publication (*Yarnykh, 2007*). Structural scans were estimated from the TRUFI volumes, with banding artifacts minimized by taking the maximum intensity across volumes (*Bangerter et al., 2004*).

For the diffusion outputs, Tensor and Ball and Two-Stick models were fit to the DW-SSFP data as described in *Tendler et al., 2020b*. In brief, fitting was performed using the full DW-SSFP Buxton model (*Buxton, 1993*), estimating a shared set of diffusion orientations (e.g., tensor eigenvectors), and a unique set of diffusivity estimates (e.g., tensor eigenvalues) per DW-SSFP flip angle. The fitting process incorporated the estimated T1, T2, and B1 maps, in addition to a noise-floor correction.

The DW-SSFP sequence does not have a well-defined b-value. To address this, the diffusivity estimates at each flip angle were combined to generate diffusivity estimates at an effective b-value of 4000 s/mm$^2$. Details of this procedure, in addition to the motivation behind the choice of 4000 s/mm$^2$ are detailed in *Tendler et al., 2020b*. Note that a small modification was made to the original minimization procedure, as described below.

**Appendix 1—table 1.** DW-SSFP Acquisition parameters at 0.5, 1.0, and 2.0 mm.

| DW-SSFP (0.5 mm) | | DW-SSFP (1.0 mm) | |
|---|---|---|---|
| q-value (cm$^{-1}$) | 300 | q-value (cm$^{-1}$) | 300 |
| Diffusion gradient duration (ms) | 14.10 | Diffusion gradient duration (ms) | 14.10 |
| Diffusion gradient strength (mTm$^{-1}$) | 50 | Diffusion gradient strength (mTm$^{-1}$) | 50 |
| Flip angles (°) | 33 and 98 | Flip angles (°) | 33 and 98 |
| No. of directions (per flip angle) | 90 | No. of directions (per flip angle) | 60 |
| No. of non-DW (per flip angle) | 6 (q=20 cm$^{-1}$) | No. of non-DW (per flip angle) | 5 (q=20 cm$^{-1}$) |

*Appendix 1—table 1 Continued on next page*

*Appendix 1—table 1 Continued*

| DW-SSFP (0.5 mm) | | DW-SSFP (1.0 mm) | |
|---|---|---|---|
| Resolution ($\mu m^3$) | 500·500·500 | Resolution ($mm^3$) | 1.0·1.0·1.0 |
| TE (ms) | 21 | TE (ms) | 21 |
| TR (ms) | 30 | TR (ms) | 30 |
| EPI factor | 1 | EPI factor | 1 |
| Bandwidth (Hz per pixel) | 198 | Bandwidth (Hz per pixel) | 130 |
| Acquisition time (per direction/non-DW) | 45 min 03 s | No. of averages | 1 |
| Acquisition time (total) | 6 days 0 hr | | |
| No. of averages | 1 | | |

| DW-SSFP (2.0 mm) | |
|---|---|
| q-value ($cm^{-1}$) | 300 |
| Diffusion gradient duration (ms) | 14.10 |
| Diffusion gradient strength ($mTm^{-1}$) | 50 |
| Flip angles (°) | 33 and 98 |
| No. of directions (per flip angle) | 221 |
| No. of non-DW (per flip angle) | 6 (q=20 $cm^{-1}$) |
| Resolution ($mm^3$) | 2.0·2.0·2.0 |
| TE (ms) | 21 |
| TR (ms) | 30 |
| EPI factor | 1 |
| Bandwidth (Hz per pixel) | 130 |
| No. of averages | 1 |

**Appendix 1—table 2.** Acquisition parameters for the TIR, TSE, structural (TRUFI), and B1-mapping (AFI) sequences.

| Turbo inversion-recovery (TIR) | | True-Fast Imaging with SSFP (TRUFI) | |
|---|---|---|---|
| Resolution ($mm^3$) | 0.75·0.75·1.60 | Resolution ($\mu m^3$) | 312.5·312.5·500 |
| Number of inversions | 6 | TE (ms) | 5.95 |
| TE (ms) | 12 | TR (ms) | 11.9 |
| TR (ms) | 1000 | Flip angle (°) | 35 |
| TIs (ms) | 31, 62, 125, 250, 500, and 850 | Bandwidth (Hz per pixel) | 130 |
| Flip angle (°) | 180 | Phase increments (°) | 0 and 180 |
| Bandwidth (Hz per pixel) | 199 | Number of averages (per set of increments) | 16 |
| Number of averages | 1 | | |

| Turbo spin-echo (TSE) – $T_2$ | | Actual flip-angle imaging (AFI) – $B_1$ | |
|---|---|---|---|
| Resolution ($mm^3$) | 0.75·0.75·1.60 | Resolution ($mm^3$) | 1.50·1.50·1.50 |
| Number of echoes | 6 | TE (ms) | 1.5 |

*Appendix 1—table 2 Continued on next page*

*Appendix 1—table 2 Continued*

| Turbo inversion-recovery (TIR) | | True-Fast Imaging with SSFP (TRUFI) | |
|---|---|---|---|
| TEs (ms) | 14, 28, 42, 56, 70, and 84 | TR$_1$/TR$_2$ (ms) | 6/30 |
| TR (ms) | 1,000 | Flip angle (°) | 60 |
| Flip angle (°) | 180 | Bandwidth (Hz per pixel) | 630 |
| Bandwidth (Hz per pixel) | 130 | Number of averages | 1 |
| Number of averages | 1 | | |

## Modification to minimization procedure

*Tendler et al., 2020b* described an approach to estimate DW-SSFP diffusivity estimates at a single effective b-value, achieved by incorporating a non-Gaussian diffusion model into the DW-SSFP signal equations. In *Tendler et al., 2020b*, non-Gaussianity was modeled using a Gamma distribution of diffusivities, estimating a mean ($D_m$) and standard deviation ($D_s$) of the Gamma distribution per voxel. Here, the Gamma fitting procedure (Equation 4 in *Tendler et al., 2020b* was replaced with:

$$\min_{D_{m_i}, D_{s_i}} \left\| \frac{\left( L_{i_{\text{sim}: \ \alpha_{\text{low}}}} (D_{m_i}, D_{s_i}) - L_{i_{\text{exp}: \ \alpha_{\text{low}}}} \right)}{SD \left( L_{i_{\text{exp}: \ \alpha_{\text{low}}}} \right)} \right\|_2^2 + \left\| \frac{\left( L_{i_{\text{sim}: \ \alpha_{\text{high}}}} (D_{m_i}, D_{s_i}) - L_{i_{\text{exp}: \ \alpha_{\text{high}}}} \right)}{SD \left( L_{i_{\text{exp}: \ \alpha_{\text{high}}}} \right)} \right\|_2^2, \tag{1}$$

where $\alpha_{\text{low}}$ and $\alpha_{\text{high}}$ are the voxelwise DW-SSFP flip angles, $L_{i_{\text{exp}: \ \alpha_{\text{low}}/\alpha_{\text{high}}}}$ are the voxelwise experimental diffusivity estimates (Tensor eigenvalues or Ball and Two-Stick diffusivity estimates) at each flip angle, $L_{i_{\text{sim}: \ \alpha_{\text{low}}/\alpha_{\text{high}}}}$ are the simulated diffusivity estimates for a given $D_{m_i}$ and $D_{s_i}$, and $SD \left( L_{i_{\text{exp}: \ \alpha_{\text{low}}/\alpha_{\text{high}}}} \right)$ are the estimated experimental standard deviation of the diffusivity estimates. This approach was found to reduce spurious diffusivity estimates in regions of low SNR. For further details of the modeling approach, see *Tendler et al., 2020b*.

## PLI preparation, scanning, and processing

Tissue samples from the anterior commissure, corpus callosum, occipital lobe gurus, pons, thalamus, and external capsule were extracted from the post-mortem brain. Samples were stored in a 30% sucrose solution with PBS and 0.025% azide for 3 weeks. Tissue blocks were subsequently embedded in optimal cutting temperature compound (Sakura, Finetek Inc, USA) and frozen to –80°C. 60-µm sections were cut from the tissue blocks with a cryostat microtome (Leica, Germany). No tissue staining was performed, as birefringence is naturally expressed by the myelin sheath.

PLI was performed using a Leica DM4000B microscope, equipped with a polarizing filter, a quarter wave plate (QWP), and a rotatable analyzer with orientation $\rho$. Samples were illuminated with a white LED (pE-100$^{\text{wht}}$ Cooled). The fast axis of the QWP was oriented 45° with respect to the transmission axis of the polarising filter to create circular polarization. The rotating analyzer captured the phase shift induced by the myelin sheath. A total of 18 images were acquired for each field of view at equidistant analyzer orientation angles, $\rho = \left\{ 0°, 10°, \dots, 170° \right\}$. Images were magnified 1.25× (0.04 NA, Leica) and captured with a Leica DFC420 CCD camera (4 µm/pixel). The green color image channel was used for further analysis.

The entire sample was imaged via raster scanning, with each row composed of multiple contiguous field-of-views (FOVs). These FOVs were automatically stitched together using in-house software (MATLAB 2015b, MathWorks, Natick, MA). For each specimen, a series of background images were acquired to correct for illumination inhomogeneities (*Dammers et al., 2010*). Microscopic fiber orientations were derived using Jones calculus (*Jones, 1941*), as described below.

## PLI fiber orientations

The light intensity ($I$) for a birefringent specimen inside a PLI-setup is described using Jones calculus (*Jones, 1941*), defining:

$$I \left( \rho \right) = \frac{I_0}{2} \left[ 1 + \sin \left( 2\rho - 2\varphi \right) \cdot \sin \delta \right], \tag{2}$$

where $I_0$ is the average light intensity, $\rho$ is the polarizer orientation, $\varphi$ is the in-plane orientation of the myelin sheath, and $\delta$ is the phase shift, defined as:

$$\delta \approx 2\pi \frac{d \cdot \Delta n}{\lambda} \cdot \cos^2 \alpha,$$

(3)

where $d$ is the sample thickness, $\Delta n$ is the sample birefringence, $\lambda$ is the light wavelength, and $\alpha$ is the inclination angle of the myelin sheath (α). Microscopic fiber orientations were derived using as above, fitting to each pixel in the raw PLI images as previously reported in *Axer et al., 2011*.

## Digital Brain Zoo datasets

Here, we provide the acquisition and processing protocol for four previously unreleased datasets to the *Digital Brain Zoo*.

### European wolf and Hamadryas baboon

Formalin-fixed European wolf and Hamadryas baboon brains were provided by Copenhagen zoo. Prior to scanning, the brains were rehydrated using a PBS solution. The size of the European wolf and Hamadryas baboon brain necessitated scanning on the Siemens 7T whole-body scanner (1 Tx/28 Rx knee coil, QED), using the brain holder displayed in *Appendix 3—figure 2*. The brain holder was filled with fluorinert (FC-3283, 3M) during the scanning procedure, a susceptibility matched fluid that gives off no signal. Diffusion-weighted volumes were acquired using DW-SSFP sequence. As highlighted in the main text, the choice of DW-SSFP was motivated by the sequences potential to simultaneously address the short $T_2$ and low diffusivity of fixed, post-mortem tissue, when limited to human scanners. A structural scan was additionally acquired using a TRUFI sequence (bSSFP), which produces high gray/white matter contrast in post-mortem tissue. Acquisition parameters are provided in *Appendix 1—table 3*.

Structural scans were formed by averaging over all 16 TRUFI datasets (root-mean sum of squares). Diffusion datasets were processed using a similar approach to the great ape datasets in *Bryant et al., 2021* and (*Roumazeilles et al., 2020*). In brief, a Gibbs ringing correction (*Kellner et al., 2016*) was applied to the diffusion and non-diffusion weighted datasets, with all coregistrations performed using FSL FLIRT (*Jenkinson and Smith, 2001*). Fitting was performed using the full DW-SSFP Buxton model (*Buxton, 1993*) adapted to incorporate Tensor and Ball and Two-Stick estimates. The fitting process incorporated estimated T1, T2, and B1 maps derived from a TIR, TSE, and AFI (*Yarnykh, 2007*) sequence acquired in the same session.

**Appendix 1—table 3.** Acquisition parameters for the DW-SSFP and structural (TRUFI) sequences. The only difference between the European wolf and Hamadryas baboon acquisition was the number of non-diffusion weighted directions acquired (13 for wolf and 11 for baboon).

| DW-SSFP | | True-Fast Imaging with SSFP (TRUFI) | |
|---|---|---|---|
| q-value (cm⁻¹) | 300 | Resolution (μm³) | 217·217·220 |
| Diffusion gradient duration (ms) | 13.56 | TE (ms) | 7.33 |
| Diffusion gradient strength (mTm⁻¹) | 52 | TR (ms) | 14.65 |
| Flip angle (°) | 39 | Flip angle (°) | 30 |
| No. of directions | 160 | Bandwidth (Hz per pixel) | 100 |
| No. of non-DW | 13/11 (q=20 cm⁻¹) | Phase increments (°) | 0, 45, 90, 135, 180, 225, 270, and 315 |
| Resolution (μm³) | 600·600·600 | No. of averages (per set of increments) | 2 |
| TE (ms) | 21 | | |
| TR (ms) | 29 | | |
| EPI factor | 1 | | |

*Appendix 1—table 3 Continued on next page*

*Appendix 1—table 3 Continued*

| DW-SSFP | True-Fast Imaging with SSFP (TRUFI) |
|---|---|
| Bandwidth (Hz per pixel) 100 | |
| Acquisition time (per direction/non-DW) | 16 min 25 s |
| Acquisition time (total) | 1 day 20 hr |
| No. of averages | 1 |

## Cotton-Top and Golden Lion tamarins

Formalin-fixed Cotton-Top and Golden Lion tamarin brains were provided by Copenhagen zoo. Prior to scanning, the brains were rehydrated using a phosphate-buffered saline solution. Scanning was performed using a 7T magnet with Agilent Direct-Drive console and 72 mm ID quadrature birdcage RF coil (Rapid Biomedical GmbH). The brain holder was filled with fluorinert during the scanning procedure, a susceptibility matched fluid that gives off no signal. Diffusion-weighted volumes were acquired using diffusion-weighted spin-echo protocol with single line readout (DW-SEMS) sequence. Acquisition parameters are provided in *Appendix 1—table 4*.

The diffusion datasets were processed using a similar approach to the prosimian and monkey data in *Bryant et al., 2021*. Datasets were preprocessed using FSL tools implemented in the Phoenix module of the MR Comparative Anatomy Toolbox (Mr Cat, https://www.neuroecologylab.org/). Tensor and ball and Two/Three-stick estimates were derived using FSL's dtifit and bedpostX (*Behrens et al., 2007*).

**Appendix 1—table 4.** Acquisition parameters for the DW-SEMS sequence.

| DW-SEMS | |
|---|---|
| b-value (s/mm$^2$) | 4000 |
| δ (ms) | 7 |
| Δ (ms) | 13 |
| Diffusion gradient strength (mTm$^{-1}$) | 320 |
| No. of directions | 128 |
| No. of non-DW | 16 |
| Resolution (μm$^3$) | 300·300·300 |
| TE (ms) | 25 |
| TR (s) | 10 |
| EPI factor | 1 |
| Bandwidth (kHz) | 100 |
| Acquisition time (per direction/non-DW) | 21 min 20 s |
| Acquisition time (total) | 2 days 4 hr |
| No. of averages | 1 |

# Appendix 2

## Human ALS MRI-histology callosum analysis

A comparison of diffusivity properties between the ALS and control cohort (12 ALS and 3 control brains) was performed in the corpus callosum, as displayed in *Figure 3c* (Main Text). To achieve this, a standard-space mask of the corpus callosum was first generated using the Jülich atlas (*Eickhoff et al., 2005*). The callosum mask was subsequently split into five distinct regions of interest (ROIs) associated with specific fiber projections as proposed by Hofer and Frahm (*Hofer and Frahm, 2006*), and transformed into the space of each post-mortem brain. Briefly, a standard space FA template (FMRIB58_FA, available as part of FSL) was modified to display similar contrast to the post-mortem FA maps. Coregistration matrices were subsequently estimated between the FA map of each post-mortem brain and the modified standard space FA template using a non-linear coregistration (ANTS) (*Avants et al., 2011*). The callosum masks were subsequently coregistered into the space of each post-mortem brain using the estimated coregistration matrices, and multiplied by a white matter mask (generated using FAST; *Zhang et al., 2001*) to remove any remaining gray matter regions.

Diffusion estimates were obtained by taking the mean over each ROI. Values were normalized to the splenium (Par/Temp/Occ) estimate, which has been proposed as a control region with little pathological burden in ALS (*Cardenas et al., 2017*). Differences in the normalized FA, MD, axial, and radial diffusivity between the ALS and control cohort were assessed with a two-tailed, family-wise error rate (FWER) corrected t-test using PALM (*Winkler et al., 2014*). Full results are provided in *Appendix 2—table 1*. Although our statistical analysis does account for sample size, it does not consider other confounds that may contribute to differences between the two groups (e.g., age and sex). Source data for the corpus callosum analysis are provided in a *Supplementary file 1*.

**Appendix 2—table 1.** p-values associated with differences between the ALS and control cohort for the diffusivity estimates.
Here 'p' defines the p-value, and '$P_{FWER}$' defines the FWER-corrected p-value ($*$=$p<0.05$; $**$=$p_{FWER}<0.05$). The largest differences between the ALS and control cohort were found in the Body (Pre/Supp Motor) category, followed by the Genu (PreFrontal) and Body (Motor) category. No differences were found in the Body (Sensory) category.

| | Body (Sensory) | Body (Motor) | Body (Pre/Supp Motor) | Genu (PreFrontal) |
|---|---|---|---|---|
| Fractional anisotropy (FA) | p=0.34 $p_{FWER}$=0.70 | p=0.042* $p_{FWER}$=0.11 | p=0.0044* $p_{FWER}$=0.013** | p=0.013* $p_{FWER}$=0.037** |
| Mean diffusivity (MD) | p=0.99 $p_{FWER}$=1.00 | p=0.18 $p_{FWER}$=0.48 | p=0.015* $p_{FWER}$=0.053 | p=0.037* $p_{FWER}$=0.12 |
| Axial diffusivity (AD) | p=0.53 $p_{FWER}$=0.92 | p=0.58 $p_{FWER}$=0.95 | p=0.084 $p_{FWER}$=0.27 | p=0.11 $p_{FWER}$=0.32 |
| Radial diffusivity (RD) | p=0.66 $p_{FWER}$=0.97 | p=0.073 $p_{FWER}$=0.23 | p=0.0022* $p_{FWER}$=0.015** | p=0.022* $p_{FWER}$=0.062 |

## Appendix 3

### Post-mortem brain holders

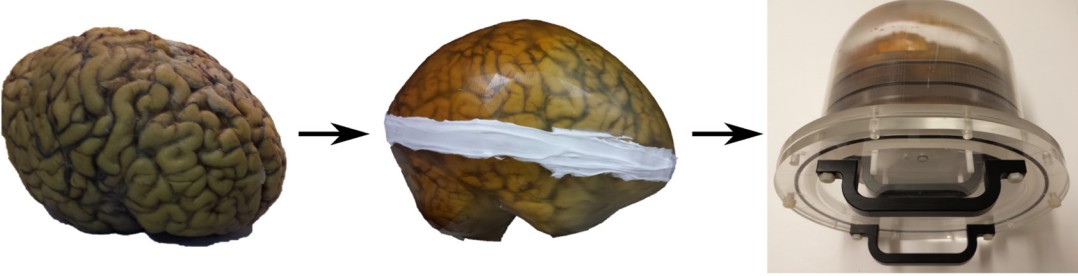

Post-mortem Brain      3D Printed Shell      Custom Holder

**Appendix 3—figure 1.** Post-mortem human brain holder. The brain holder ensures consistent placement during scanning. Here, the custom holder tightly seals the brain in place, whilst the 3D printed shell (provided by Dr Alard Roebroeck, Maastricht University) prevents pressure on the brain. The holder is designed with a spherical cavity to maximize field homogeneity. For further information on our scanning procedure for human post-mortem brains, see *Wang et al., 2020*.

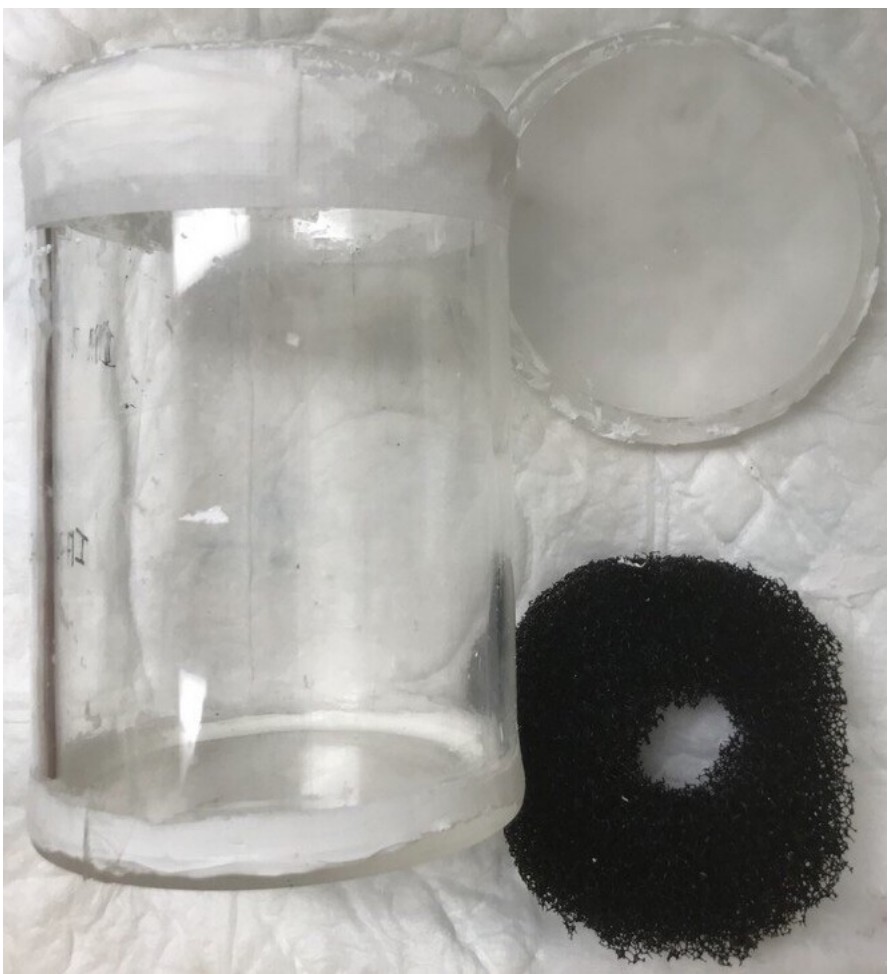

**Appendix 3—figure 2.** Post-mortem cylindrical brain holder. The brain holder used for scanning large nonhuman brains which fit inside the 28 channel QED knee coil. This consisted of a cylindrical container, with plastic gauze (black) used to secure samples during the acquisition.

## Appendix 4

### Example structural MRI dataset

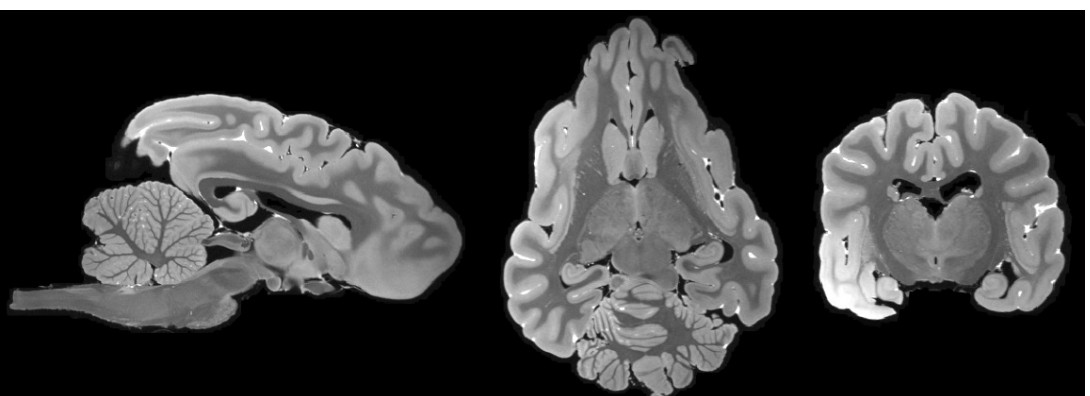

**Appendix 4—figure 1.** Structural MRI. Example structural MRI dataset acquired using a bSSFP sequence in the European wolf (*Canis lupus*) at a resolution of 220 μm (isotropic). bSSFP Structural MRI datasets display excellent gray-white matter contrast, facilitating the delineation of fine tissue structures and integration with processing pipelines for surface reconstruction. Contrast in bSSFP datasets is reversed compared to conventional T1-weighted structural MRI scans (gray matter appears bright, and white matter appears dark), which must be accounted for in any analysis pipeline.

