## [Editor Report]

This paper describes a new open-access digital brain bank of post-mortem brains that have been scanned with high-resolution, multimodal magnetic resonance imaging and with select datasets accompanied by histological data. This valuable resource can be used to study healthy human brains, pathological human brains, and the brains of other species, opening new opportunities for comparative neuroanatomy and the biological validation of non-invasive neuroimaging signals.

---

## [Decision Letter]

**Decision letter after peer review:**

Thank you for submitting your article "The Digital Brain Bank, an open access platform for post-mortem datasets" for consideration by *eLife*. Your article has been reviewed by 3 peer reviewers, and the evaluation has been overseen by a Reviewing Editor and Christian Büchel as the Senior Editor. The following individuals involved in review of your submission have agreed to reveal their identity: Ilona Lipp (Reviewer #1); Konrad Wagstyl (Reviewer #2); Timo Dickscheid (Reviewer #3).

Essential revisions:

1. Please modify the term "interactive data discovery and release platform" which, as used in the abstract, a little bit misleading. The interactivity is limited to viewing overlays of different images in a few of the datasets and the release option of one's own datasets is not established (yet).

2. Please add the resolution of all scans to Table 1.

3. Regarding the sentence "Although MRI hardware and acquisition protocols often need to be tailored to a specific domain, the signals measured in all these settings are fundamentally the same (Boon et al., 2019)." -> Please clarify the intention of this statement--is this an argument for using post-mortem MRI?

4. Please clarify: in Table 1, when it says structural MRI, is this always a T1w image?

5. Please clarify: From the text, it is not clear what diffusion data are made available exactly. Is this all diffusion-weighted images or just the estimated parameters, or does this depend on the database? ("To facilitate cross-dataset comparisons, the majority of diffusion datasets from the Digital Brain Bank provide derived diffusivity estimates in the form of diffusion tensor and/or ball and sticks model parameters (Behrens et al., 2007)). To reflect this, we primarily provide curated datasets to facilitate these analyses, as opposed to outputs associated with the results of specific projects (e.g. tractography-derived maps)…. Curated unprocessed data is additionally provided when there are clear routes of investigation (e.g. diffusion MRI volumes to investigate alternative diffusion models)."

6. Please clarify what you mean by 'tractography-derived maps' and 'clear routes of investigation'? (P.S.: "data are" not "data is")

7. The authors have chosen here to present the MRI data co-registered to the 2D histology and not vice versa. Sectioning can introduce morphological shifts in the tissue that might alter neuroanatomical findings relative to the original brain structure. Please add a note to explain why they chose to present the registration this way round.

8. The paper puts much emphasis on describing the postmortem MRI aquisition details, but the abstract suggests a clear focus on the neuroinformatics aspect. Please update the abstract to describe details about the postmortem acquisition component.

9. It would be very helpful if the publications that are listed for datasets were actual hyperlinks, so one could directly access them and need not copy-paste them in the browser.

10. Please specify how version control will be managed for future datasets. Given the nature of the data, more high-resolution scans might be added to existing datasets. How would users of the platform keep track of this?

11. Please place the paragraph on the Tensor Image Registration Library in the methods section rather than in the results.

*Reviewer #1 (Recommendations for the authors):*

It would help adding the resolution of all scans to Table 1, especially since the advantages of high resolution were so strongly advertised with Figure 1.

Regarding the sentence "Although MRI hardware and acquisition protocols often need to be tailored to a specific domain, the signals measured in all these settings are fundamentally the same (Boon et al., 2019)." -> I was not sure what you are trying to say here, is this an argument for using post-mortem MRI? If yes, this is not super clear from what comes before.

Table 1 very helpful, when it says structural MRI, is this always a T1w image?

From the text, it is not clear to me what diffusion data are made available exactly, is this all weighted images or just the estimated parameters, or does this depend on the database? ("To facilitate cross-dataset comparisons, the majority of diffusion datasets from the Digital Brain Bank provide derived diffusivity estimates in the form of diffusion tensor and/or ball and sticks model parameters (Behrens et al., 2007))…. To reflect this, we primarily provide curated datasets to facilitate these analyses, as opposed to outputs associated with the results of specific projects (e.g. tractography-derived maps)…. Curated unprocessed data is additionally provided when there are clear routes of investigation (e.g. diffusion MRI volumes to investigate alternative diffusion models)."

Also, could you please clarify what you mean by 'tractography-derived maps' and 'clear routes of investigation'? (P.S.: "data are" not "data is")

*Reviewer #2 (Recommendations for the authors):*

- It is currently unclear in table 1 which species have both structural and diffusion scans. These could be indicated with a symbol such as an asterisk.

*Reviewer #3 (Recommendations for the authors):*

– The paper puts much emphasis on describing the postmortem MRI aquisition details, but the abstract suggested to me a clear focus on the neuroinformatics aspect. I think this additional focus would be beneficial to point out more clearly in the abstract already.

– To me, the platform would benefit a lot from a visual (3D) overview illustrating where in each whole-brain dataset the additional microscopy datasets have been measured. This could even be a pre-computed cross-sectional view with markers.

– Figure 3: The caption "the digital pathologist" suggests to see the corresponding software interface, but the figure rather illustrates typical content.

– It would be very helpful if the publications that are listed for datasets were actual hyperlinks, so one could directly access them and need not copy-paste them in the browser.

– Do you foresee versioning of the datasets? Given the nature of the data, I imagine that e.g. more high-resolution scans might be added to existing datasets. How would users of the platform keep track of this?

– l. 525 ff: I would expect the paragraph on the Tensor Image Registration Library rather in the methods section than in the results.

[Editors' note: further revisions were suggested prior to acceptance, as described below.]

Thank you for resubmitting your work entitled "The Digital Brain Bank, an open access platform for post-mortem datasets" for further consideration by *eLife*. Your revised article has been evaluated by Christian Büchel (Senior Editor) and a Reviewing Editor.

The manuscript has been improved but there are some remaining issues that need to be addressed, as outlined below:

– Re: R1 C1:

"established laboratory models" may sound like disease models, which is intuitive for the digital pathologist, but a bit restrictive for the digital anatomist. Please rephrase.

– Re: R1 C8:

"The signals measured in all these settings are fundamentally the same" is somewhat vague, as due to the same technology, the type of signal measured is the same, however, how to interpret and utilise the signal is quite different, which makes the "direct translation between domains" challenging. This paragraph of the manuscript can be slightly misleading. Please rephrase.

– in vivo is not spelled consistently across the manuscript (e.g. lines 75 and 77). Please amend

– The discussion with reviewer 3 on version control was important. While not raised in the initial round of reviewer comments, please consider adding DOIs to individual datasets. Typically, when datasets are shared through repositories they are assigned dataset DOIs (e.g. figshare, zenodo, University repos), which are essential for reproducible data science. Currently users of this platform can cite DOIs for the dataset's paper of origin and this platform. Perhaps within the future goals for version control, DOIs for specific version might also be added.

*Reviewer #1 (Recommendations for the authors):*

The authors have provided thorough explanations and amendments in response to my previously raised comments. I think that the manuscript has substantially improved now in terms of clarity. I have no more major comments.

*Reviewer #2 (Recommendations for the authors):*

The authors have responded in detail to the reviewer comments. The responses to my initial comments were addressed and overall the manuscript and description of the resource is improved.

*Reviewer #3 (Recommendations for the authors):*

Thank you for the detailed responses to the remarks on the first manuscript.

All my concerns have been clarified in detail in the author's response, and appropriately addressed in the revised text. Especially by toning down the platform aspect and putting more emphasis on the system as a data resource, I find the manuscript now very consistent.

From my perspective, it can be accepted in the present form.

---

## [Author Response]

Essential revisions:1. Please modify the term "interactive data discovery and release platform" which, as used in the abstract, a little bit misleading. The interactivity is limited to viewing overlays of different images in a few of the datasets and the release option of one's own datasets is not established (yet).

To address this comment, we have replaced the term “interactive data discovery and release platform” with “data release platform” (Page 2 Line 41, Page 3 Line 87). We hope this reduces the anticipated functionality on the website and puts full emphasis on the focus of the Digital Brain Bank as a data resource.

Importantly, the Digital Brain Bank is not currently intended to be a general release platform for post-mortem datasets. Rather, it is a platform for release of datasets acquired/curated by researchers at the University of Oxford and close collaborators. We will additionally consider Derived Outputs (Defined in the response to Comment 5) from users of Digital Brain Bank datasets, subject to quality control on an individual basis. This ensures we can main a high-level of quality control throughout.

To ensure this is more clearly communicated, we have added the following text to the manuscript:

Introduction (Page 3 Lines 104-106)

“All datasets uploaded to the Digital Brain Bank are associated with researchers at the University of Oxford, or from close collaborators. Limited Derived Outputs from users of Digital Brain Bank datasets will also be considered for data upload.”

Methods (Subsection: Conditions for Data Uploading) (Page 17 Lines 642-645)

“All datasets hosted on the Digital Brain Bank are associated with projects performed at the University of Oxford, in collaboration with members of the University of Oxford, or from close collaborators. In addition, limited Derived Outputs from users of Digital Brain Bank datasets will also be considered for data upload, subject to quality control on an individual basis.”

where Derived Outputs is defined in the response to comment **5** and the *Methods* section.

2. Please add the resolution of all scans to Table 1.

The resolution of all scans has been added to Table 1 (Page 5)

3. Regarding the sentence "Although MRI hardware and acquisition protocols often need to be tailored to a specific domain, the signals measured in all these settings are fundamentally the same (Boon et al., 2019)." -> Please clarify the intention of this statement--is this an argument for using post-mortem MRI?

To help frame our response, we begin by pasting in the original paragraph that included this statement (Page 3 Lines 57-67):

“Magnetic resonance imaging (MRI) occupies a unique position in the neuroscience toolkit. In humans, MRI is used at the single subject level diagnostically, and is increasingly deployed at the population level in epidemiology (Marcus et al., 2007; Miller et al., 2016; Snoek et al., 2021; Van Essen et al., 2013). MRI is well-established in the context of imaging causal manipulations in experimental organisms ranging from mice (Denic et al., 2011; Thiessen et al., 2013) to non-human primates (Absinta et al., 2017; Klink et al., 2021), and provides precise measurements in cellular and tissue preparations (Wilhelm et al., 2012). This extensive landscape of overlap with the broader neuroscience toolkit creates the potential for MRI to facilitate integration between technologies and investigations. Although MRI hardware and acquisition protocols often need to be tailored to a specific domain, the signals measured in all these settings are fundamentally the same (Boon et al., 2019). There are few methods available to neuroscientists that span this breadth of domains.”

As noted at the beginning of the paragraph, MRI can be utilised in many investigative domains spanning a broad range of species and in conjunction with many different experimental techniques. In all cases, the fundamental measurement (signal) is the same. Nevertheless, to optimise imaging in these different domains, we typically make alterations that amount to differences in protocol: we may use an MRI scanner with specialised hardware, or an image acquisition technique optimised for different tissue properties. Although there are subtleties that often must be addressed in analysis to account for these differences in protocol, the fact that the fundamental nature of the measurement is the same greatly facilitates the comparison of datasets. Taken together, this enables us to directly translate findings across domains.

To clarify this point in the manuscript, we have made two changes. Firstly, we have removed the reference to Boon et al. Whilst this paper does demonstrate a clear example of utilising the same MRI scanner to investigate different domains (in vivo and post-mortem), we appreciate that it shifts the sentences focus onto post-mortem imaging only. Secondly, we have modified the sentence to (Page 3 Lines 64-66):

“Although MRI hardware and acquisition protocols often need to be tailored to a specific domain, the signals measured in all these settings are fundamentally the same, facilitating cross-domain comparisons.”

We hope that this small change succinctly emphasises the utility of MRI, whilst generalising the focus away from post-mortem imaging.

4. Please clarify: in Table 1, when it says structural MRI, is this always a T1w image?

Please clarify: in Table 1, when it says structural MRI, is this always a T1w image?

Structural MRI scans available in the first release to the Digital Brain Bank are not derived from T1w images. Rather, they are all derived from balanced SSFP sequences, with the exception of two structural scans (1x Thylacine and 1x Tasmanian devil) acquired using a T2-weighted acquisition. The motivation behind this is provided in the Discussion (Subsection: Structural MRI) (Pages 13 Lines 461-469):

“Structural MRI enables the delineation of fine tissue structures and cortical surface reconstruction through high contrast, high-resolution imaging datasets. However, convergence of T1 relaxation times for grey and white matter in formalin fixed post-mortem tissue leads poor contrast with conventional T1-weighted structural protocols (Miller et al., 2011). All structural MRI available in the first data release were acquired using either a balanced SSFP (bSSFP) or T2-weighted sequence, which demonstrate excellent grey/white matter contrast in fixed post-mortem tissue. Notably, bSSFP signal forming mechanisms lead to an extremely high SNR-efficiency (even when considering the reduced T1 and T2 of post-mortem tissue), affording the acquisition of ultra-high resolution (< 500 μm) imaging volumes to delineate fine tissue structures in large post-mortem samples.”

To clarify this in the manuscript, the following text has been added to the image caption in Table 1 (Page 5):

“All Structural MRI datasets in the first release were acquired using a balanced SSFP (bSSFP) or T2-weighted sequence, which yields excellent grey-white matter contrast in fixed post-mortem tissue. Diffusion MRI datasets were acquired using a combination of diffusion-weighted steady-state free precession (DW-SSFP) and diffusion-weighted spin-echo (DW-SE) sequences. Full details of the motivation behind the choice of sequences and available contrasts are described in the Discussion.”

Like conventional T1w acquisitions, contrast in bSSFP and T2w structural MRI datasets is predominantly driven by grey and white matter. A key distinction is that grey/white matter contrast is reversed (grey matter appears bright, and white matter appears dark). Our work with these datasets has demonstrated that they can be incorporated with conventional structural MRI analysis pipelines, provided the reversal of image contrast is accounted for.

To address this, we have added the following text to the Discussion (Subsection: Structural MRI) (Page 14 Lines 471-476):

“Contrast in bSSFP and T2-weighted structural MRI datasets is reversed in comparison to conventional in vivo T1-weighted acquisitions (i.e., grey matter appears bright, and white matter appears dark). For these datasets, image contrast is predominantly driven by grey and white matter, facilitating the delineation of fine tissue structures and surface reconstructions (Roumazeilles et al., 2020). An example bSSFP dataset is displayed in Appendix 4 Figure 1. Integration with conventional structural MRI processing pipelines often needs to account for the reversal of image contrast.”

5. Please clarify: From the text, it is not clear what diffusion data are made available exactly. Is this all diffusion-weighted images or just the estimated parameters, or does this depend on the database? ("To facilitate cross-dataset comparisons, the majority of diffusion datasets from the Digital Brain Bank provide derived diffusivity estimates in the form of diffusion tensor and/or ball and sticks model parameters (Behrens et al., 2007)). To reflect this, we primarily provide curated datasets to facilitate these analyses, as opposed to outputs associated with the results of specific projects (e.g. tractography-derived maps)…. Curated unprocessed data is additionally provided when there are clear routes of investigation (e.g. diffusion MRI volumes to investigate alternative diffusion models)."

Data availability

Thank you for the feedback that our original manuscript was not clear when distinguishing between datasets at different stages of the analysis pipeline. To address this in the revised manuscript, we have added the following text to the Methods (Subsection: Types of Data Provided) (Page 17 Lines 653-672):

“The Digital Brain Bank uses the following definitions:

Raw Data: MRI and microscopy images obtained directly from the MRI or slide scanner. These data require further processing be useful (i.e. the value in individual pixels is not directly informative about biological processes or properties).Pre-Processed Outputs: Imaging outputs from individual brains produced by a first stage of processing suitable for a broad range of subsequent analysis. For example, diffusion parameter estimates (e.g. Diffusion Tensor outputs), quantitative relaxometry maps, or PLI fiber orientation maps. These outputs can be immediately fed into an analysis to answer a hypothesis-or data-driven neuroscientific question.Derived Outputs: Results produced from subsequent analysis that use Pre-Processed Outputs as an input specific to a domain of neuroscientific investigation. For example, tractography-derived pathway segmentations or group-averaged atlases.

The Digital Brain Bank aims to capture a broad range of datasets under the umbrella of post-mortem neuroimaging, and as such we’ve aimed to keep the resource flexible for uploaded datasets. There are no strict criteria regarding the types and structure of post-mortem data released to the Digital Brain Bank. In addition, the Digital Brain Bank aims to facilitate the investigation of research hypotheses designed by the user. To reflect this, we primarily provide Pre-Processed Outputs. Limited Derived Outputs associated with specific projects will be also made available at the discretion of the Digital Brain Bank. Raw data for a given dataset is available on request.”

Diffusion MRI data

Provided diffusion MRI Pre-Processed Outputs (e.g. diffusion tensor estimates) depends on the individual dataset. Notably, there are no widely available software packages that incorporate the correct signal model for diffusion-weighted steady-state free precession (DW-SSFP). We share our custom software for analysing these data through the Digital Brain Bank. To clarify this, we have modified the following text to the Methods (Subsection: Types of Data Provided) (Page 17 Lines 679-686):

“For diffusion MRI datasets acquired with DW-SE, we also provide the Pre-Processed Outputs of individual diffusion volumes, which users can feed into a broad range of available software. For DW-SSFP, we do not by default provide the individual diffusion volumes, as no widely available diffusion MRI software packages incorporate the DW-SSFP signal model. We share our custom software for analysing these data through the Digital Brain Bank. DW-SSFP Diffusion volumes are available upon request, with the caveat that care needs to be taken in analysing these data in light of the unusual dependences (e.g., T1 and T2) and signal model of DW-SSFP (Buxton 1993; Tendler, Foxley, Cottaar, et al., 2020; Tendler, Foxley, Hernandez-Fernandez, et al., 2020).”

6. Please clarify what you mean by 'tractography-derived maps' and 'clear routes of investigation'? (P.S.: "data are" not "data is")

Following the above response to comment 5, the referenced sentences referring to ‘tractography-derived maps’ and ‘clear routes of investigation’ have been removed from the manuscript.

All instances of ‘data is’ have been corrected in the manuscript

7. The authors have chosen here to present the MRI data co-registered to the 2D histology and not vice versa. Sectioning can introduce morphological shifts in the tissue that might alter neuroanatomical findings relative to the original brain structure. Please add a note to explain why they chose to present the registration this way round.

All histology stains acquired in the first release to the Digital Brain Bank consist of a single 2D slice per given region. Co-registering a single slice of histology data into the MRI space transforms a 2D surface (histology) into a 3D volume (MRI). Due to the cutting angle and non-linear deformations during tissue sampling and sectioning, the 2D histology surface will pass through multiple MRI voxel planes. Examining these data in the 3D MRI space is challenging, particularly when considering the high-resolution of the histology data. Thus, we have chosen to examine our MRI-histology coregistrations in the histology space only.

Regarding the morphological shifts, the MRI-histology coregistration software designed by co-author Istvan Huszar (TIRL) has been specifically designed to deal with deformations induced during cutting. Our TIRL pipeline has a specific stage for estimating these deformations from photographs and MRI, and a later stage that refines those deformations for the specific histology slice. For further information, please refer to the original TIRL manuscript *(Huszar et al., 2019).*

To address this in the manuscript, we have expanded our description of TIRL in the Methods (Subsection: Tensor Image Registration Library (TIRL)) (Pages 16 Lines 631-638):

“The decision to present MRI-microscopy coregistrations in the 2D histology space (Figure 3d) was chosen to facilitate visualisation. During manual histology sampling, the cutting process introduces non-linear deformities. Furthermore, the cutting angle is not constrained to be parallel to the MRI voxel plane. In general, the excised tissue used for histology will pass through multiple MRI voxel planes non-linearly, limiting visualisation in the 3D MRI space. Importantly, TIRL has been specifically designed to deal with the deformations induced during cutting. Our TIRL pipeline has a specific stage for estimating these deformations from photographs and MRI, and a later stage that refines those deformations for the specific histology slice. Further information is provided in (Huszar et al., 2019).”

8. The paper puts much emphasis on describing the postmortem MRI aquisition details, but the abstract suggests a clear focus on the neuroinformatics aspect. Please update the abstract to describe details about the postmortem acquisition component.

The abstract has been updated to describe the post-mortem acquisition component, and reads as follows (Page 2):

“Post-mortem MRI provides the opportunity to acquire high-resolution datasets to investigate neuroanatomy, and validate the origins of image contrast through microscopy comparisons. We introduce the Digital Brain Bank (open.win.ox.ac.uk/DigitalBrainBank), a data release platform providing open access to curated, multimodal post-mortem neuroimaging datasets. Datasets span three themes – Digital Neuroanatomist: datasets for detailed neuroanatomical investigations; Digital Brain Zoo: datasets for comparative neuroanatomy; Digital Pathologist: datasets for neuropathology investigations. The first Digital Brain Bank release includes twenty-one distinctive whole-brain diffusion MRI datasets for structural connectivity investigations, alongside microscopy and complementary MRI modalities. This includes one of the highest-resolution whole-brain human diffusion MRI datasets ever acquired, whole-brain diffusion MRI in fourteen non-human primate species, and one of the largest post-mortem whole-brain cohort imaging studies in neurodegeneration. The Digital Brain Bank is the culmination of our lab’s investment into post-mortem MRI methodology and MRI-microscopy analysis techniques. This manuscript provides a detailed overview of our work with post-mortem imaging to date, including the development of diffusion MRI methods to image large post-mortem samples, including whole, human brains. Taken together, the Digital Brain Bank provides cross-scale, cross-species datasets facilitating the incorporation of post-mortem data into neuroimaging studies.”

9. It would be very helpful if the publications that are listed for datasets were actual hyperlinks, so one could directly access them and need not copy-paste them in the browser.

This has been updated throughout the *Digital Brain Bank* website

10. Please specify how version control will be managed for future datasets. Given the nature of the data, more high-resolution scans might be added to existing datasets. How would users of the platform keep track of this?

In the longer term, we agree that more formalised version control will be necessary to manage datasets and streamline communicating updates without human intervention. Integration of version control is a future ambition of the *Digital Brain Bank*.

A key reason why version control has not yet been considered is that at present, data access is achieved by contacting the Request Data Contact on the *Digital Brain Bank* website via email, followed by signing MTAs with the University of Oxford prior to manual data transfer. Given the ad hoc nature of this data access, we propose that the best current approach for platform users to keep track of changes is to maintain lists of users who have requested access to individual datasets. Users will be contacted when any changes are made.

We plan on streamlining user sign-up, authentication, and approval on the *Digital Brain Bank* website. This will enable approved users to access datasets directly on the website. Within this updated system, we propose to keep track of a dataset by associating it with a version number. This number will be updated when new components are added or modifications are made to existing files, and subsequently directly communicated to data users via their user account.

To communicate the status of data access more clearly and our proposed future plans, we have added the following subsection to the *Methods* (Page 18 Lines 705-717). This text incorporates a further comment from Reviewer 3 regarding programmatic interactions:

“Data Storage Database

As is the nature of a data resource associated with both completed and ongoing projects, some datasets will be updated over time. These future releases will typically be associated with new images being made available or improvements to existing processing pipelines. Until more streamlined data access methods are in place, we will contact users directly to inform them of any updates made to a given dataset. This approach is aligned with the current framework for data access, with users required to contact the Request Data Contact on the Digital Brain Bank website to request access.

A future ambition of the Digital Brain Bank is to streamline data access procedures by integrating user sign-up, authentication, and approval combined with access to specific dataset versions within our database over time. To achieve this, we are continuously developing the platform and incorporating feedback and feature requests including enabling a programmatic interface to datasets for approved users, and detailed dataset versioning.”

11. Please place the paragraph on the Tensor Image Registration Library in the methods section rather than in the results.

This paragraph has been moved to the Methods section (Subsection: Tensor Image Registration Library (TIRL)) (Pages 16- Lines 622-638)

Reviewer #1 (Recommendations for the authors):It would help adding the resolution of all scans to Table 1, especially since the advantages of high resolution were so strongly advertised with Figure 1.

The resolution of all scans has been added to Table 1 (Page 5)

Regarding the sentence "Although MRI hardware and acquisition protocols often need to be tailored to a specific domain, the signals measured in all these settings are fundamentally the same (Boon et al., 2019)." -> I was not sure what you are trying to say here, is this an argument for using post-mortem MRI? If yes, this is not super clear from what comes before.

To help frame our response, we begin by pasting in the original paragraph that included this statement (Page 3 Lines 57-67):

“Magnetic resonance imaging (MRI) occupies a unique position in the neuroscience toolkit. In humans, MRI is used at the single subject level diagnostically, and is increasingly deployed at the population level in epidemiology (Marcus et al., 2007; Miller et al., 2016; Snoek et al., 2021; Van Essen et al., 2013). MRI is well-established in the context of imaging causal manipulations in experimental organisms ranging from mice (Denic et al., 2011; Thiessen et al., 2013) to non-human primates (Absinta et al., 2017; Klink et al., 2021), and provides precise measurements in cellular and tissue preparations (Wilhelm et al., 2012). This extensive landscape of overlap with the broader neuroscience toolkit creates the potential for MRI to facilitate integration between technologies and investigations. Although MRI hardware and acquisition protocols often need to be tailored to a specific domain, the signals measured in all these settings are fundamentally the same (Boon et al., 2019). There are few methods available to neuroscientists that span this breadth of domains.”

As noted at the beginning of the paragraph, MRI can be utilised in many investigative domains spanning a broad range of species and in conjunction with many different experimental techniques. In all cases, the fundamental measurement (signal) is the same. Nevertheless, to optimise imaging in these different domains, we typically make alterations that amount to differences in protocol: we may use an MRI scanner with specialised hardware, or an image acquisition technique optimised for different tissue properties. Although there are subtleties that often must be addressed in analysis to account for these differences in protocol, the fact that the fundamental nature of the measurement is the same greatly facilitates the comparison of datasets. Taken together, this enables us to directly translate findings across domains.

To clarify this point in the manuscript, we have made two changes. Firstly, we have removed the reference to Boon et al. Whilst this paper does demonstrate a clear example of utilising the same MRI scanner to investigate different domains (in vivo and post-mortem), we appreciate that it shifts the sentences focus onto post-mortem imaging only. Secondly, we have modified the sentence to (Page 3 Lines 64-66):

“Although MRI hardware and acquisition protocols often need to be tailored to a specific domain, the signals measured in all these settings are fundamentally the same, facilitating cross-domain comparisons.”

We hope that this small change succinctly emphasises the utility of MRI, whilst generalising the focus away from post-mortem imaging.

Table 1 very helpful, when it says structural MRI, is this always a T1w image?

Structural MRI scans available in the first release to the Digital Brain Bank are not derived from T1w images. Rather, they are all derived from balanced SSFP sequences, with the exception of two structural scans (1x Thylacine and 1x Tasmanian devil) acquired using a T2-weighted acquisition. The motivation behind this is provided in the Discussion (Subsection: Structural MRI) (Page 13 Lines 461-469):

“Structural MRI enables the delineation of fine tissue structures and cortical surface reconstruction through high contrast, high-resolution imaging datasets. However, convergence of T1 relaxation times for grey and white matter in formalin fixed post-mortem tissue leads poor contrast with conventional T1-weighted structural protocols (Miller et al., 2011). All structural MRI available in the first data release were acquired using either a balanced SSFP (bSSFP) or T2-weighted sequence, which demonstrate excellent grey/white matter contrast in fixed post-mortem tissue. Notably, bSSFP signal forming mechanisms lead to an extremely high SNR-efficiency (even when considering the reduced T1 and T2 of post-mortem tissue), affording the acquisition of ultra-high resolution (< 500 μm) imaging volumes to delineate fine tissue structures in large post-mortem samples.”

To clarify this in the manuscript, the following text has been added to the image caption in Table 1 (Page 5):

“All Structural MRI datasets in the first release were acquired using a balanced SSFP (bSSFP) or T2-weighted sequence, which yields excellent grey-white matter contrast in fixed post-mortem tissue. Diffusion MRI datasets were acquired using a combination of diffusion-weighted steady-state free precession (DW-SSFP) and diffusion-weighted spin-echo (DW-SE) sequences. Full details of the motivation behind the choice of sequences and available contrasts are described in the Discussion.”

Like conventional T1w acquisitions, contrast in bSSFP and T2w structural MRI datasets is predominantly driven by grey and white matter. A key distinction is that grey/white matter contrast is reversed (grey matter appears bright, and white matter appears dark). Our work with these datasets has demonstrated that they can be incorporated with conventional structural MRI analysis pipelines, provided the reversal of image contrast is accounted for.

To address this, we have added the following text to the Discussion (Subsection: Structural MRI) (Page 14 Lines 471-476):

“Contrast in bSSFP and T2-weighted structural MRI datasets is reversed in comparison to conventional in vivo T1-weighted acquisitions (i.e., grey matter appears bright, and white matter appears dark). For these datasets, image contrast is predominantly driven by grey and white matter, facilitating the delineation of fine tissue structures and surface reconstructions (Roumazeilles et al., 2020). An example bSSFP dataset is displayed in Appendix 4 Figure 1. Integration with conventional structural MRI processing pipelines often needs to account for the reversal of image contrast.”

From the text, it is not clear to me what diffusion data are made available exactly, is this all weighted images or just the estimated parameters, or does this depend on the database? ("To facilitate cross-dataset comparisons, the majority of diffusion datasets from the Digital Brain Bank provide derived diffusivity estimates in the form of diffusion tensor and/or ball and sticks model parameters (Behrens et al., 2007))…. To reflect this, we primarily provide curated datasets to facilitate these analyses, as opposed to outputs associated with the results of specific projects (e.g. tractography-derived maps)…. Curated unprocessed data is additionally provided when there are clear routes of investigation (e.g. diffusion MRI volumes to investigate alternative diffusion models)."

Data availability

Thank you for the feedback that our original manuscript was not clear when distinguishing between datasets at different stages of the analysis pipeline. To address this in the revised manuscript, we have added the following text to the *Methods* (Subsection: *Types of Data Provided*) (Page 17 Lines 653-672):

“The Digital Brain Bank uses the following definitions:

Raw Data: MRI and microscopy images obtained directly from the MRI or slide scanner. These data require further processing be useful (i.e. the value in individual pixels is not directly informative about biological processes or properties).Pre-Processed Outputs: Imaging outputs from individual brains produced by a first stage of processing suitable for a broad range of subsequent analysis. For example, diffusion parameter estimates (e.g. Diffusion Tensor outputs), quantitative relaxometry maps, or PLI fiber orientation maps. These outputs can be immediately fed into an analysis to answer a hypothesis-or data-driven neuroscientific question.Derived Outputs: Results produced from subsequent analysis that use Pre-Processed Outputs as an input specific to a domain of neuroscientific investigation. For example, tractography-derived pathway segmentations or group-averaged atlases.

The Digital Brain Bank aims to capture a broad range of datasets under the umbrella of post-mortem neuroimaging, and as such we’ve aimed to keep the resource flexible for uploaded datasets. There are no strict criteria regarding the types and structure of post-mortem data released to the Digital Brain Bank. In addition, the Digital Brain Bank aims to facilitate the investigation of research hypotheses designed by the user. To reflect this, we primarily provide Pre-Processed Outputs. Limited Derived Outputs associated with specific projects will be also made available at the discretion of the Digital Brain Bank. Raw data for a given dataset is available on request.”

Diffusion MRI data

Provided diffusion MRI Pre-Processed Outputs (e.g. diffusion tensor estimates) depends on the individual dataset. Notably, there are no widely available software packages that incorporate the correct signal model for diffusion-weighted steady-state free precession (DW-SSFP). We share our custom software for analysing these data through the Digital Brain Bank. To clarify this, we have modified the following text to the Methods (Subsection: Types of Data Provided) (Page 17 Lines 679-686):

“For diffusion MRI datasets acquired with DW-SE, we also provide the Pre-Processed Outputs of individual diffusion volumes, which users can feed into a broad range of available software. For DW-SSFP, we do not by default provide the individual diffusion volumes, as no widely available diffusion MRI software packages incorporate the DW-SSFP signal model. We share our custom software for analysing these data through the Digital Brain Bank. DW-SSFP Diffusion volumes are available upon request, with the caveat that care needs to be taken in analysing these data in light of the unusual dependences (e.g., T1 and T2) and signal model of DW-SSFP (Buxton 1993; Tendler, Foxley, Cottaar, et al., 2020; Tendler, Foxley, Hernandez-Fernandez, et al., 2020).”

Also, could you please clarify what you mean by 'tractography-derived maps' and 'clear routes of investigation'? (P.S.: "data are" not "data is")

Following the above response, the referenced sentences referring to ‘tractography-derived maps’ and ‘clear routes of investigation’ have been removed from the manuscript.

All instances of ‘data is’ have been corrected in the manuscript

Reviewer #2 (Recommendations for the authors):- It is currently unclear in table 1 which species have both structural and diffusion scans. These could be indicated with a symbol such as an asterisk.

This has been added to Table 1 (Page 5)

Reviewer #3 (Recommendations for the authors):– The paper puts much emphasis on describing the postmortem MRI aquisition details, but the abstract suggested to me a clear focus on the neuroinformatics aspect. I think this additional focus would be beneficial to point out more clearly in the abstract already.

The abstract has been updated to describe the post-mortem acquisition component, and reads as follows (Page 2):

“Post-mortem MRI provides the opportunity to acquire high-resolution datasets to investigate neuroanatomy, and validate the origins of image contrast through microscopy comparisons. We introduce the Digital Brain Bank (open.win.ox.ac.uk/DigitalBrainBank), a data release platform providing open access to curated, multimodal post-mortem neuroimaging datasets. Datasets span three themes – Digital Neuroanatomist: datasets for detailed neuroanatomical investigations; Digital Brain Zoo: datasets for comparative neuroanatomy; Digital Pathologist: datasets for neuropathology investigations. The first Digital Brain Bank release includes twenty-one distinctive whole-brain diffusion MRI datasets for structural connectivity investigations, alongside microscopy and complementary MRI modalities. This includes one of the highest-resolution whole-brain human diffusion MRI datasets ever acquired, whole-brain diffusion MRI in fourteen non-human primate species, and one of the largest post-mortem whole-brain cohort imaging studies in neurodegeneration. The Digital Brain Bank is the culmination of our lab’s investment into post-mortem MRI methodology and MRI-microscopy analysis techniques. This manuscript provides a detailed overview of our work with post-mortem imaging to date, including the development of diffusion MRI methods to image large post-mortem samples, including whole, human brains. Taken together, the Digital Brain Bank provides cross-scale, cross-species datasets facilitating the incorporation of post-mortem data into neuroimaging studies.”

– To me, the platform would benefit a lot from a visual (3D) overview illustrating where in each whole-brain dataset the additional microscopy datasets have been measured. This could even be a pre-computed cross-sectional view with markers.

It is helpful to know that users might appreciate this additional information. It is not clear that *Tview* as conceived is the right tool to provide this information. Instead, and as mentioned noted in the response to Reviewer 3, we are currently integrating a second web-based viewer to the *Digital Brain Bank* website, *NiiVue*. Niivue is being actively developed by second author Taylor Hanayik, and provides web-based visualisation and interaction of 3D NIFTI volumes. This viewer offers two advantages: Firstly, it enables navigation of 3D MRI datasets directly on the Digital Brain Bank website prior to downloading data. Secondly, it provides the option of overlays, which we will use to localise the histology sampling location in a 3D dataset.

A NiiVue demonstration (with overlays) is available at: https://niivue.github.io/niivue/features/overlay.multiplanar.html. We anticipate that this viewer will greatly improve the degree of interaction available on the *Digital Brain Bank* website.

To address this in the manuscript, we have added the following text in the *Discussion* (Page 15 Lines 565-570):

“Future Directions: Dataset Visualisation

To improve visualisation of MRI-only datasets on the Digital Brain Bank website, we are currently integrating NiiVue (Rorden et al., 2021), a web-based 3D volumetric viewer for navigating MRI datasets. NiiVue additionally supports binary overlays, which will be used to visualise the location of tissue sampling in the brain. Further details are available at https://github.com/niivue/niivue.”

*(Rorden et al., 2021)*: niivue/niivue: 0.13.0 (0.13.0). Zenodo. https://doi.org/10.5281/zenodo.5786270

– Figure 3: The caption "the digital pathologist" suggests to see the corresponding software interface, but the figure rather illustrates typical content.

The three figures in the manuscript all provide an overview of the typical content available for each of the three research themes, rather than details of the software interface.

– It would be very helpful if the publications that are listed for datasets were actual hyperlinks, so one could directly access them and need not copy-paste them in the browser.

This has been updated throughout the *Digital Brain Bank* website

– Do you foresee versioning of the datasets? Given the nature of the data, I imagine that e.g. more high-resolution scans might be added to existing datasets. How would users of the platform keep track of this?

In the longer term, we agree that more formalised version control will be necessary to manage datasets and streamline communicating updates without human intervention. Integration of version control is a future ambition of the *Digital Brain Bank*.

A key reason why version control has not yet been considered is that at present, data access is achieved by contacting the Request Data Contact on the *Digital Brain Bank* website via email, followed by signing MTAs with the University of Oxford prior to manual data transfer. Given the ad hoc nature of this data access, we propose that the best current approach for platform users to keep track of changes is to maintain lists of users who have requested access to individual datasets. Users will be contacted when any changes are made.

As mentioned above, we plan on streamlining user sign-up, authentication, and approval on the *Digital Brain Bank* website. This will enable approved users to access datasets directly on the website. Within this updated system, we propose to keep track of a dataset by associating it with a version number. This number will be updated when new components are added or modifications are made to existing files, and subsequently directly communicated to data users via their user account.

To communicate the status of data access more clearly and our proposed future plans, we have added the following subsection to the *Methods*:

“Data Storage Database

As is the nature of a data resource associated with both completed and ongoing projects, some datasets will be updated over time. These future releases will typically be associated with new images being made available or improvements to existing processing pipelines. Until more streamlined data access methods are in place, we will contact users directly to inform them of any updates made to a given dataset. This approach is aligned with the current framework for data access, with users required to contact the Request Data Contact on the Digital Brain Bank website to request access.

A future ambition of the Digital Brain Bank is to streamline data access procedures by integrating user sign-up, authentication, and approval combined with access to specific dataset versions within our database over time. To achieve this, we are continuously developing the platform and incorporating feedback and feature requests including enabling a programmatic interface to datasets for approved users, and detailed dataset versioning.”

– l. 525 ff: I would expect the paragraph on the Tensor Image Registration Library rather in the methods section than in the results.

This paragraph has been moved to the *Methods* section (Subsection: *Tensor Image Registration Library (TIRL)*) (Pages 16 Lines 622-638)

[Editors' note: further revisions were suggested prior to acceptance, as described below.]

The manuscript has been improved but there are some remaining issues that need to be addressed, as outlined below:– Re: R1 C1:"established laboratory models" may sound like disease models, which is intuitive for the digital pathologist, but a bit restrictive for the digital anatomist. Please rephrase.

The sentence has been updated as follows (Page 4 Line 133):

“Datasets within the Digital Anatomist provide a new direction for answering fundamental questions in neuroanatomy, through ultra-high resolution MRI data and complementary microscopy within the same sample in humans and model non-human species.”

– Re: R1 C8:"The signals measured in all these settings are fundamentally the same" is somewhat vague, as due to the same technology, the type of signal measured is the same, however, how to interpret and utilise the signal is quite different, which makes the "direct translation between domains" challenging. This paragraph of the manuscript can be slightly misleading. Please rephrase.

We certainly do not want to suggest that translating across-domains is trivial using MRI. To address this, we have removed the phrase “fundamentally the same”, we appreciate that readers could interpret this as relating to signal interpretation and utilisation. The sentence has been rewritten to clarify that cross-domain comparisons are facilitated by the common underlying *technology* associated with all MRI measurements as follows (Page 3 Lines 64-66):

“Although MRI hardware and acquisition protocols often need to be tailored to a specific domain, the underlying technology associated with all MRI measurements gives rise to a common set of signal forming mechanisms, facilitating cross-domain comparisons.”

– in vivo is not spelled consistently across the manuscript (e.g. lines 75 and 77). Please amend

This has been updated throughout the manuscript.

– The discussion with reviewer 3 on version control was important. While not raised in the initial round of reviewer comments, please consider adding DOIs to individual datasets. Typically, when datasets are shared through repositories they are assigned dataset DOIs (e.g. figshare, zenodo, University repos), which are essential for reproducible data science. Currently users of this platform can cite DOIs for the dataset's paper of origin and this platform. Perhaps within the future goals for version control, DOIs for specific version might also be added.

Thank you for this comment – we agree that directly associating DOIs with a given dataset would be a valuable addition to the Digital Brain Bank resource. Given the current approach of data access and design of the Digital Brain Bank repository, DOIs are not currently supported for individual datasets. However, this is a future ambition for the resource, which will be investigated amongst the other points raised in the previous comments to Reviewer 3 surrounding version control. We will assess the options for associating DOIs directly through the Digital Brain Bank resource, or via existing providers as raised in your original comment. To reflect this in the manuscript, we have updated the text in the Methods (subsection: Data Storage Database) as follows (Page 15 Lines 718-720):

“A future ambition of the Digital Brain Bank is to streamline data access procedures by integrating user sign-up, authentication, and approval combined with access to specific dataset versions within our database over time. To achieve this, we are continuously developing the platform and incorporating feedback and feature requests including enabling a programmatic interface to datasets for approved users, and detailed dataset versioning. We will investigate associating dataset versions with DOIs directly on the Digital Brain Bank website (or through known providers), to facilitate the tracking and reproducibility of individual datasets and analysis pipelines.”